# Perspectives on the origin of language: Infants vocalize most during independent vocal play but produce their most speech-like vocalizations during turn taking

**Helen L. Long** [1]*, **Gordon Ramsay** [2,3,4], **Ulrike Griebel** [5,6,7], **Edina R. Bene** [6], **Dale D. Bowman** [5,8], **Megan M. Burkhardt-Reed** [6], **D. Kimbrough Oller** [5,6,7]

1 Waisman Center, University of Wisconsin-Madison, Madison, Wisconsin, United States of America, 2 Marcus Autism Center, Children's Healthcare of Atlanta, Atlanta, Georgia, United States of America, 3 Department of Pediatrics, Emory University School of Medicine, Atlanta, Georgia, United States of America, 4 Center for Translational Social Neuroscience, Emory University, Atlanta, Georgia, United States of America, 5 Institute for Intelligent Systems, University of Memphis, Memphis, Tennessee, United States of America, 6 School of Communication Sciences and Disorders, University of Memphis, Memphis, Tennessee, United States of America, 7 Konrad Lorenz Institute for Evolution and Cognition Research, Klosterneuburg, Austria, 8 Department of Mathematics, University of Memphis, Memphis, Tennessee, United States of America

* helen.long@wisc.edu

**Data Availability Statement:** Data and materials associated with this manuscript are available at https://osf.io/ery6b/.

## Abstract

A growing body of research emphasizes both endogenous and social motivations in human vocal development. Our own efforts seek to establish an evolutionary and developmental perspective on the existence and usage of speech-like vocalizations ("protophones") in the first year of life. We evaluated the relative occurrence of protophones in 40 typically developing infants across the second-half year based on longitudinal all-day recordings. Infants showed strong endogenous motivation to vocalize, producing vastly more protophones during independent vocal exploration and play than during vocal turn taking. Both periods of vocal play and periods of turn-taking corresponded to elevated levels of the most advanced protophones (canonical babbling) relative to periods without vocal play or without turn-taking. Notably, periods of turn taking showed even more canonical babbling than periods of vocal play. We conclude that endogenous motivation drives infants' tendencies to explore and display a great number of speech-like vocalizations, but that social interaction drives the production of the most speech-like forms. The results inform our previously published proposal that the human infant has been naturally selected to explore protophone production and that the exploratory inclination in our hominin ancestors formed a foundation for language.

## Introduction

The emergence of the human vocal capacity beginning in infancy has long been the subject of inquiry and speculation [1–4]. By the second half of the last century, empirical findings from

**Funding:** National Institute on Deafness and Other Communication Disorders, R01DC015108, Dr. D. Kimbrough Oller; National Institute of Mental Health, P50MH100029, Dr. Gordon Ramsay; National Institute of Child Health and Human Development, T32HD007489, Dr. Helen L. Long; National Institute of Child Health and Human Development, U54HD090256, Dr. Helen L. Long; Plough Foundation of Memphis, Dr. D. Kimbrough Oller; the Holly Lane Foundation, Dr. Gordon Ramsay; the Marcus Foundation, Dr. Gordon Ramsay; the Woodruff-Whitehead Foundation, Dr. Gordon Ramsay; the Georgia Research Alliance, Dr. Gordon Ramsay.

**Competing interests:** The authors have declared that no competing interests exist.

experimental as well as quantitative observational studies were being reported, focused heavily on vocal interaction between caregivers and infants [5–9]. That focus has continued, with scores of articles yearly reporting evidence of effects of face-to-face engagement on the learning of language-like skills and the acquisition of phonology [10–17]. The traditional viewpoint has been extended to include a widespread belief in and research to substantiate the claim that phonological forms are learned in the first year by infant imitation [18–22]. The expectation that imitation drives early learning of speech-like sounds is robustly echoed in the literature on other babbling species (e.g., songbirds), where parallels with human babbling are often drawn [23–26].

A contrasting viewpoint, represented by a smaller number of publications, has existed for many years and is being expanded in recent time. This literature, reviewed by Vihman [27], cites a longstanding recognition of the existence of endogenous vocalization in infancy and suggests that although vocal interaction plays an important role in the development of speech, much if not most of the vocal activity of human infants is neither imitative nor even socially interactive. Findings that are being further emphasized recently indicate that immediate vocal imitation occurs rarely, estimated as including not more than ~5% of vocal events in human infants during the first year, and there is reason to believe the amount may be considerably lower, given the difficulty of discerning who is imitating whom in sequences where infants and mothers interchange similar sounds [28–30]. It is well established based on recent work that mothers vocally imitate infants far more than the reverse, but even mothers imitate only a small minority of infant sounds in face-to-face interaction [28].

Perhaps even more importantly, new research has indicated that the great majority of infant speech-like vocalizations are directed to no one. It has been shown that typically developing infants produce such vocalizations predominantly endogenously, that is, with no indication of social directivity [31]. Even in laboratory recordings during periods when parents were instructed to seek social interaction with infants, most speech-like vocalizations (~60%) were judged as not being directed to them. This predominance of endogenous vocalization was even stronger (~80%) when parents were present with infants but not attempting to engage them.

Other research from our laboratories using human-coding of randomly sampled segments from all-day recordings has revealed that human infant speech-like vocalization occurs at prodigious rates. We refer to these infant speech-like sounds as "protophones" [32], sometimes termed "babbling," a usage that leaves open confusion since it has often been used to apply to canonical babbling only. We, on the other hand, use the term protophone to encompass early noncanonical forms such as vowel-like sounds ("vocants"), squeals, and growls, as well as later-occurring canonical babbling, which includes syllables sounding like, for example, "wa", "baba", or "dada". Infant volubility has been estimated in these studies to involve on average ~ 3500 protophones per day, 4–5 per minute every waking hour from the first month of life, a rate that vastly outstrips the rate of crying and fussing even from the first 3 months [33, 34]. Additional observations have even raised the possibility that infants vocalize as much or more when they are alone as when they are in the presence of possible interactors [35, 36]. Furthermore, it is now well-documented that even profoundly deaf infants produce protophones in the first year at rates that are comparable to the rates of hearing infants [37, 38].

The seemingly implicit assumption in the bulk of the child development literature then, that infant vocalizations are routinely produced under the influence of vocal interaction, has thus been challenged directly by empirical evidence, and the countervailing trend appears to be strengthening, focusing increasingly on a role for intrinsic motivation in the development of emotional and cognitive systems, including those related to protophones [39–42]. Because protophones are overwhelmingly produced in the absence of social interaction and often when

infants are alone in a room, it is hard to escape the impression that these vocalizations are internally motivated, constituting a special kind of exploratory play that occurs very frequently in human infants, but rarely if ever in other ape infants [41].

Play is well established as a feature of development throughout infancy [42–47]. Stark [48] described vocal play as highly variable, with infants producing sounds in new and repeated combinations, modifying patterns and features during bouts of independent infant vocal activity. During vocal play, we see infants exploring the vocal apparatus and seemingly practicing various properties of sounds such as amplitude, pitch, and syllabic structure.

While it is true that endogenous infant vocalization has been recognized for decades [49, 50], it is only very recently that quantitative comparisons of endogenous and interactive vocalization rates have been attempted. The present work will carry that quantification further, focusing on endogenous and interactive vocalizations in all-day recordings in the homes of infants.

## On the evolution of human infant vocal patterns

Why is the human infant so inclined to explore vocalization? We and (independently) John L. Locke have proposed an "ultimate" answer, in the terminology of Tinbergen [51], an answer involving reflections on ancient hominin history. The work is intended to posit evolutionary conditions that fostered the emergence of vocal foundations for language, long before language existed. The proposal suggests that during the evolution of our species, humans came to be endowed with a tendency to take pleasure in playful, exploratory vocalization [52–55]. The reasoning relies in part on the altriciality (that is, the relative helplessness) of the hominin infant, a condition believed to have been caused by the narrowing of the hominin pelvis owing to our obligate bipedalism originating in ancient times and the consequent requirement that the schedule of development be revised to afford a smaller head at birth [56–58]. The whole term of development was thus apparently slowed in ancient hominins, yielding infants with greater need for long-term caregiver investment than in related species. In modern times the development of human infants to the point of self-provisioning is proportionally about twice as long as in other apes [59–61].

As the reasoning goes, the altricial hominin infant thus came under selective pressure to provide new kinds of fitness signals to caregivers, who then offered selective care to those infants whose signals were most effective. The vocal system, being in a sense largely free of other utilitarian requirements (unlike the hands, which are regularly used for manipulation and grasping), became a particular target for selection in response to altriciality. And thus, if the reasoning is correct, hominins became vocal fitness signalers, at first in infancy, but as evolution proceeded, the vocal system was modified to have far greater complexity and power and came to be used at all stages of life, routinely as a fitness signal, but also (and simultaneously during speech acts) as a tool for social engagement, planning, group coordination, and so on. According to the reasoning, language became possible eventually after additional capabilities evolved, but it has always been dependent upon the vocal flexibility that was the product of copious vocal fitness signaling evolved first in hominin infants.

This hypothesis is the only proposal on the table, as far as we know, to account for the initial split of hominins from their ape relatives in terms of vocal activity and vocal communication. The need for language cannot have supplied selection pressure for that split because language did not exist at that point. All naturally-selected changes must be accounted for in terms of pressures available at the time of the change and cannot be referred to possibilities that might occur in the future [62–65]. We have argued extensively that without the inclination and ability to produce vocalization flexibly, vocal language could not have begun to be evolved in the

hominin line, just as we have argued that vocal language could not be developed in modern human infants without the vocal exploratory tendency—all acts of human language require vocal flexibility [55, 66–68].

It can, of course, be reasoned that the existence of language, as it began to emerge across hominin evolution could also have resulted, presumably gradually, in selective pressure on infant vocalization as an early language-like exercise, in addition to the pressures that yielded vocal fitness signaling. Thus there is reason to suspect that nowadays infant vocal activity and practice even in the first months of life does indeed support the development of language [69, 70]. This does not contradict the existence of an even deeper inclination for exploratory vocalization in the modern infant that was selected in ancient times. Our reasoning is thus compatible with at least two factors, one supported by adaptation for vocal fitness-signaling and one supplying preparation for language, in the emergence and maintenance human infant vocal tendencies.

The fitness-signaling hypothesis offers a reason, we have proposed, that human infants seem to enjoy exploratory vocal play. They appear to have been endowed with a tendency to treat the voice in much the way all primates treat playful activity with the hands. No other ape shows the tendency to produce purely playful vocalization to our knowledge [41], and yet the human infant tends to produce exploratory vocalization every day at prodigious rates.

But how often does this playful activity coincide with or occur independently from vocal turn-taking? The availability of all-day recording technology opens the door to more representative sampling to assess this question quantitatively by evaluating the relative amounts of endogenous vocal production and vocal turn taking in human infants.

## Canonical babbling in vocal play and social interaction

The reasoning about the fitness-signaling hypothesis responds to Tinbergen's *ultimate* questions about why capabilities and inclinations exist (the two questions regarding evolution and survival value), but it does not respond to Tinbergen's *proximate* questions (the two questions regarding development and causation mechanism) [51]. There is a well-developed and growing literature on the proximate questions that takes particular note of the emergence of increasingly advanced protophones across the first two years of life [27, 69–71].

A key indicator of especially advanced protophone production, and conceivably a particularly salient fitness signal, is canonical babbling, which consists of syllables that are often so well-formed that they can constitute speech if used in the right circumstances [30, 72]. Canonical babbling has been long established as a robust stage of prelinguistic vocal development, occurring prior to the emergence of early words, and constituting a necessary foundation for extensive vocabulary development [73–75]. To our knowledge, there is no published research evaluating the role of exploratory motivation in infants' production of canonical babbling and no direct evaluation of the extent to which social engagement in vocal turn taking affects it. In the present research, we observed canonical babbling in naturalistic settings recorded longitudinally for infants in the first year. Segments extracted from all-day home audio recordings were rated for levels of infant independent exploratory vocal play and vocal turn taking, allowing measurement of the degree of social and non-social vocal activity and thus, exploratory and social motivations, respectively. The research offers the opportunity to assess not only the quantity, but also the *quality* of infant vocalizations as reflected in canonical babbling syllables as a proportion of all protophone syllables.

In line with the fitness-signaling hypothesis, it seems reasonable to expect for infants to produce their most advanced vocal forms during periods of maximal caregiver attention, i.e.,

during social interaction. Empirical evidence has been presented that caregivers are keenly aware of their infants' developmental capabilities, including their vocal sophistication [76–78]. Higher rates of canonical syllables (as opposed to less advanced protophones) during social interaction suggests a motivation to display advanced protophones when caregivers are attentive. Furthermore, it seems reasonable to expect high levels of exploratory vocal play to be accompanied by particularly high levels of canonical babbling because exploratory activity reveals infant attention to development of a sound system, and because the attention of caregivers is surely drawn to the quality of protophones during periods where the infant is particularly active vocally.

## Testable propositions

Our study then addresses infant vocalization in both social and nonsocial circumstances, observed naturalistically in all-day recordings, which have made it possible during more than a decade to assess vocal patterns with greater ecological validity than was possible in the past [79–83]. The data were coded in 5-min segments drawn from the recordings. In one kind of analysis, we assessed the extent to which production of protophones appeared to be driven endogenously, as opposed to being elicited by caregivers. The dependent variables for this kind of analysis were 1) human coder estimates of the amount of protophone production during each 5-minute segment that constituted independent, presumably endogenous vocalization, which we shall refer to as *vocal play*, or "VP" [48], and 2) the amount of protophone production in the same 5-minute segments that constituted vocal interaction with caregivers, which we shall refer to as *turn taking*, or "TT".

To be clear, the VP and TT categories were not exclusive within coded segments; some VP and some TT could occur in the same segments. Only vocalizations deemed to be protophones for the present work (squeals, vocants, or growls) were judged to constitute either VP or TT; other less frequently occurring protophones such as non-phonated sounds or ingressive sounds were not included. Importantly, some of the vocalizations deemed to be protophones included canonical syllables and some did not. Also, VP and TT were judged by coders to occur only in cases where the recognized protophones were judged to be prominent (loud and/or long) enough that a nearby caregiver might have noticed them. Further, protophones judged to consist of whining (interpreted as "complaints") were not allowed to be treated as VP. These and other coding constraints are elaborated below under Coding Categories.

In additional analyses, the dependent variable was the frequency of occurrence of advanced infant vocal forms, as reflected in the *canonical babbling ratio* (CBR), the number of canonical syllables divided by the number of all protophone syllables, during VP and TT. Evidence has been presented suggesting that social motivation drives the production of canonical babbling and that infant-directed speech promotes it [84–86], but there has been no research to our knowledge quantifying the relative extent of canonical babbling in VP and TT for infants in the natural environment of the home.

The research focused on data at three ages during the second half-year of life (7.5, 9.5, and 12 mo). Earlier ages were not addressed since earlier protophones are almost entirely non-canonical and cannot yield useful information about CBR nor its relation to VP or TT.

**Comparison of vocal play and turn taking.** We know of no previous research that has addressed relative rates of VP and TT based on all-day recordings. We suspect that the tendency for infants to engage in VP as opposed to TT may be especially high in the naturalistic environment of the home as opposed to its occurrence in laboratory recordings [31]. Consequently, we propose:

**Proposition 1: At all Ages, coded ratings will indicate infants produce VP more often than TT**.

To our knowledge, there has been no attempt to directly evaluate a possible role for endogenous VP on the extent of infant canonical babbling. We reason that there may be a tendency of infants to explore their most speech-like utterance capabilities when they are most actively involved in VP, times during which they may be most attentive to their own vocal product, stimulating awareness of categorical growth and calibrating their vocal capacities. Further, if, as we suppose, infant protophones have been naturally selected as fitness signals, periods of VP may tend to attract attention from caregivers, the persons who apply the selection pressure on the nature and growth of infant vocal patterns according to the fitness-signaling hypothesis. Thus, we predict:

**Proposition 2: Higher CBRs will occur during recorded segments with Some VP compared to those with No VP**.

Prior research has also suggested that caregiver interaction influences infants to produce canonical babbling, suggesting an important effect of caregiver interaction on the growth of the speech capacity [84, 87]. There is also evidence that the amount of speech-related vocalization is stimulated by vocal interaction, evidence relying on the LENA algorithms applied to all-day recordings [85]; but the LENA analyses, based on automated categorizations, have not addressed canonical babbling specifically, only differentiating speech-related vocalization from other sounds such as cries or vegetative sounds. So far, to our knowledge, there has been no attempt to directly test a possible influence of TT on canonical babbling based on samples from all-day recordings. The naturalistic circumstance in the home is the most representative one to allow testing the expectation that interactions with parents actually stimulate infants to display and perhaps develop their most advanced vocal forms. It makes sense that if protophones are fitness signals [52, 55], human infants may have been naturally selected to produce more advanced vocal forms when caregivers are presumably paying attention to them. Thus, we predict:

**Proposition 3: Higher infant CBRs will occur during recording segments with Some TT compared to those with No TT**.

Finally, it seems likely that if the fitness-signaling hypothesis is on target, infants should be motivated to generate their most speech-like vocalizations when caregivers are most attentive, that is, during the circumstance of vocal interaction. Independent of the fitness-signaling hypothesis, the human infant shows many signs of being socially motivated, and consequently during vocal interaction, it makes sense that they would display their most advanced vocal forms. Thus, we make a prediction that to our knowledge has never been considered in prior research:

**Proposition 4: Higher infant CBRs will occur during recording segments with Some TT compared to Some VP**.

## Materials and methods

The institutional review boards of the University of Memphis and Emory University approved the procedures used in this study. Families provided written consent prior to participation.

### Participants

As part of an NIH-funded Autism Center of Excellence conducted at the Marcus Autism Center in Atlanta, Georgia (in affiliation with Emory University), more than 300 families of newborn infants were recruited via flyers, advertisements, social media and community referrals to participate in a longitudinal sibling study of development across the first three years of life.

Infants were recruited as being either at high risk (HR) or low risk (LR) for autism. They were deemed HR if they had at least one older biological sibling with a confirmed autism diagnosis, and LR if they had no familial history of autism in 1st, 2nd, or 3rd degree relatives.

The present study focuses on LR infants only, selected among those for whom all the data relevant to the present study have been coded completely and evaluated for coder agreement and for whom no clinical features (e.g., autism, language delay, subthreshold ASD) were detected at 24- or 36-month evaluations. The sample includes 40 LR infants, with 9 LR infants having been excluded because there was suspicion of some developmental anomaly at the time of the final evaluation. Thus, although the broader study focused on autism risk, the present analysis focuses entirely on a sample of verified typically developing infants.

In such observational research, there are many potentially influential variables that cannot practically be controlled or evaluated quantitatively. But two demographic variables for which we have relevant data are of potential interest: socio-economic status (SES) and sex. We conducted precursor analyses on these to provide additional perspective and to provide a rationale for simplifying our analyses by excluding them. Sex and self-reported maternal education (as a proxy for SES) measures were balanced in subject recruitment to the extent possible. Low/high SES groups were based on a median split of maternal education in the entire cohort (range 13 to 25 years, median 18 years). The sample for the present study was somewhat biased toward males (23 of 40) and toward higher SES (27 of 40). The latter pattern is common in longitudinal research because families of higher SES tend to volunteer for and maintain enrollment in longitudinal research more frequently.

Both SES groups and both sex groups exhibited very similar patterns as the whole group with regard to the relations between VP and TT (all far higher VP than TT) and with regard to the CBRs as a function of VP and TT (see S1 Text for details and additional comparisons). Therefore, we do not include maternal education/SES or sex as independent variables in the analyses below, thus simplifying the analyses.

## Audio recordings

Families were asked to complete audio recordings once a month from 1–36 months of age. The present study used data collected between 6.5 and 13 months of age to represent the typical range of expected onset for and high infant activity in canonical babbling. These data were grouped into three age ranges for analysis and are labeled in the tables and figures below with reference hereafter to the approximate mean age within each group: 6.5–8.49 months ("~7.5 months"), 8.5–10.49 months ("~9.5 months"), and 10.5–13 months ("~12 months").

Audio recordings were completed using battery powered LENA recording devices [80] secured inside the pocket of a special vest or clothing item with button clasps. The devices can record up to 16 hours of audio per charge. They have a 16 kHz sampling rate and given the low mouth-to-microphone distance (~10 cm), usually offer excellent audio quality for human coding of recorded material.

Once a month, parents were provided with a LENA recording device and were supplied regularly with appropriately sized clothing for their infant to wear throughout the day, as well as full instructions on how to carry out recordings. The device was returned to the research project staff at the Marcus Autism Center each month following recording days for data processing. Each family completed ~5 total recordings (range: 3–7) across the ages studied, with an average recording time of approximately 11 hours per day.

We look forward to a time, hopefully not too far into the future, when all-day recordings with both audio and video can be made available for such research. Since we assume both audition and vision form the basis for caregiver judgments about infant wellness based on

vocalization, we also assume video plus audio will offer a preferable basis for coding all the relevant features of infant vocalization.

## Coders and segment selection

The coders were 15 graduate students in the School of Communication Sciences and Disorders at the University of Memphis. They were all trained for coding as specified below. Each of the students was semi-randomly assigned to a set of infants (including all their longitudinal recordings) from among the first 100 recorded in the broader study of infants at risk and not at risk for autism, but the coders were never given diagnostic or clinical information about the infants. The number of recordings coded by the individual coders varied widely (from 5 to 22).

In accord with the standard protocol for all-day recordings in the Origin of Language Laboratories (OLL) of the University of Memphis, where all the human coding was conducted, twenty-one 5-minute segments were randomly extracted from each of the 196 all-day recordings, for coding in "Phase 1" of infant utterances. Following coding of each segment, coders completed a questionnaire to obtain information about the infant's environment (see S3 Text). After Phase 1 coding was complete, the software (Action Analysis, Coding and Training, AACT) used in the OLL [88] automatically selected eight of the 21 segments from each recording on the basis of the coded information and questionnaire responses. The 8 were selected based on several criteria: 1) relatively high infant volubility of infant utterance coding in every case, 2) in some cases based on relatively high infant-directed speech (IDS, as judged during the Phase 1 questionnaire), 3) in other cases based on infants being alone or not engaged in vocal interaction (again according to the Phase 1 questionnaire), and 4) in the remainder of the cases were selected based on maximum volubility alone. The segment selection algorithm for "Phase 2" coding is described in detail in S6 Text. These 8 segments were then coded in Phase 2 for infant syllable and IDS counts (further described in *Coding Categories* below), with another questionnaire following coding of each segment.

The final coded dataset for the present study on the 40 typically developing infants at three ages thus consisted of 1,576 five-min segments (mean = 39.4 segments/infant across the three ages; mean = 13.4 segments/infant at 7.5 mo, mean = 12.4 segments/infant at 9.5 mo, mean = 14.8 segments/infant at 12 mo) extracted from the all-day recordings. For analyses below, the number of segments was actually 1,572, eliminating 4 segments where the infants produced no protophones at all.

The coding outcomes for these segments combined from Phase 1 and Phase 2 yielded the following numbers of the present study's most important coded categories, which are defined in the next section: >57,000 infant protophones, including >108,000 infant syllables, of which >13,000 were deemed to be canonical syllables; other speakers produced >18,000 utterances directed to the infants according to the coders and >50,000 utterances not directed to the infants.

One might ask if the relatively high volubility (7.4 protophones per min) of the 8 selected segments per recording might bias the sample to show higher VP than the larger set of 21 randomly selected segments per recording, which had lower volubility (4.9 protophones per minute). In fact, the correlation between volubility and VP (some VP vs no VP) was negative in the selected sample ($\rho$ = -.022) while the correlation between volubility and TT (some TT vs no TT) was positive ($\rho$ = .132). The pattern seems to contradict the suggestion that amount of VP could be overestimated with respect to amount of TT because of the selection of relatively high volubility segments. One might also suggest that a difference in volubility might explain any difference found between CBR in segments with some VP as opposed to some TT. Indeed,

volubility was higher in segments with some TT (8.4 protophones per min) than in segments with some VP (7.3 protophones per min). Volubility could then be thought to drive a tendency for high CBR in TT if canonical babbling is indeed associated with volubility.

## Coding categories

One might imagine the best approach would be automated analysis of the whole recording set, obviating the need for random selection of a subset of recording segments and bypassing human coding. But there are no existing automated analyses, to our knowledge, that differentiate reliably among the categories of interest. In all cases, anyway, the gold standard for evaluation of automated methods is human coding. The very goal of automated analysis of infant vocalizations is to mimic human perceptions of speech-like quality.

The categories for human coding of infant vocalizations utilized in the present work are described in detail in the supplementary material to prior work from our laboratory [33], where spectrographic displays are presented to help illustrate the definitions, which are provided in S2 Text. These categories are utilized widely for research in the OLL at the University of Memphis, where all the present coding was conducted. Here, we summarize the definitions for the categories of interest in the present work.

Infant "utterances" are defined as breath-groups of phonation [89]. Thus, inhalation, or a pause perceived to leave room for inhalation, is treated as a boundary between utterances. "Fixed signals," that is, cries, whimpers, and laughs are coded, but are distinguished from the protophones, the presumed precursors to speech, which include most prominently the phonatory categories of squeals, growls, and vowel-like sounds (vocants), and far less frequently occurring sounds (whispers, raspberries, ingressive sounds, clicks, etc.). It is important to acknowledge that ~15% of protophones are perceived auditorily as manifesting negative affect [90]. We refer to these protophones as "whining", which is distinct in our scheme from crying or whimpering. A questionnaire item (see below) provides an estimate from coders on how much whining (protophones manifesting complaint) occurs in each segment.

Protophones occurring at very low intensity—so low that coders judge them to be below the normal threshold of caregiver attention—are also not coded. The distinction between cries and whimpers (both of which are treated as naturally-selected signals of negativity or distress) is formal in this research and is defined in detail, along with spectrographic exemplars, in prior studies from our laboratory [33, 91].

Vegetative utterances (burps, coughs, sneezes, effort grunts, etc.) are not coded. In our system "grunts" are always required to have a sharp glottal onset accompanying some kind of movement or strain that involves a glottal hold and diaphragmatic tension to impose rigidity on the internal torso—the effort grunt occurs at the release of the glottal hold, and is therefore considered an artifact of this movement. We do not use the term "grunt" in our coding system to refer to utterances called "attention grunts" or "communicative grunts" in a widely recognized body of work on early vocal communication by McCune et al. [92]. Instead, such sounds were coded as "vocants" for the current work. It should be noted however that only a subcategory of vocants, termed "quasivowels," see [32], would satisfy the definitions of "attention grunts" or "communicative grunts" in McCune et al.'s work.

During Phase 1, protophones, cries, whimpers and laughs are coded in real-time for each segment, as they are in the vast majority of OLL projects. During Phase 2 coding, each infant syllable within any protophone is coded as canonical or non-canonical, again in real-time. The notion "syllable" is based on perceived rhythmic beats in infant utterances, counted in a way that is analogous to the auditorily-based counting of syllables in adult speech. The notion of the canonical syllable has been utilized widely in research on human infancy since its

introduction more than four decades ago [72, 74, 93, 94]. Roughly, a canonical syllable consists of at least one well-formed, supraglottally articulated consonant-like element and at least one well-formed nucleus or vowel-like element (e.g., ma, da, etc.). Reduplicated examples of canonical babbling sequences, among the most salient indicators of infants' having reached the "canonical stage," may be heard, for example, as "baba" or "nana." Non-canonical syllables are typically composed of vowel-like nuclei only. Utterances can, of course, consist of multiple syllables.

Caregiver or other speaker "utterances" are bounded differently from infant utterances during OLL coding, since grown-ups often produce long phonated sequences without inhalation, much longer than the protophones of infants. Hence, the breath-group rule used for infant utterances is replaced for caregiver utterances with a different rule. A perceived pause or perceived sentential intonational boundary is treated as a boundary between caregiver utterances. Caregiver utterances and other-speaker utterances are categorized in real-time during Phase 2 as "infant-directed" (that is, directed to the infant wearing the recorder) or "other-directed." The distinction is usually easy to make because of the common use of "baby register" [95] or "motherese" [96] in speaking to infants and because the semantics of caregiver utterances usually makes it abundantly clear whether an infant is being addressed.

In both Phase 1 and Phase 2, coders respond to questionnaires immediately after coding each segment. There are 34 questions in all, 17 for each phase (see S3 Text). The questions of most relevance for the present work are: Phase 1, question 1 on infant-directed speech (IDS): "Did any other person talk to the baby?" Phase 2, question 1 on whining: "Were any of the infant's protophones used to complain?" Phase 2, question 3 on VP: "Were any of the infant's protophones purely vocal play or vocal exploration (not social, not trying to get something, etc.)?" And Phase 2, question 4 on TT: "Were any of the infant's protophones used in vocal turn taking with another speaker?" Detailed instructions accompanying the questions are provided in a formal instructions protocol (see S3 Text).

The questionnaire items of interest in both Phase 1 and Phase 2 invoke a 5-point Likert-scale for response to the relevant questions, e.g., for IDS, "Did any other person talk to the baby?" The scale aligns to the following designations: *1 = Never, 2 = Less than half the time, 3 = About half the time, 4 = More than half the time, 5 = Close to the whole time.* For example, a TT rating of 5 is to be applied to segments where a caregiver is judged to be speaking to the infant and the infant to be responding with protophones in back-and-forth vocal interaction throughout the whole 5-minute segment (although no rating of 5 was *ever* assigned to a segment with regard to TT in the data for the present paper). Segments with a VP rating of 5, likewise is to be applied to segments where the coder perceived the vast majority of infant protophones across the segment to be playful and/or exploratory and not directed to another person (a rating of 5 was *not* uncommon for VP). The subjective Likert Scale response system is intended to simulate the kinds of judgments caregivers make (we presume often unconsciously) regarding the relative rate of occurrence of various functions of vocalizations produced by infants throughout the day.

*Vocal Play* (VP) is defined, as inspired by Stark [48], to refer to infant protophones produced independent of social interaction or any attempt on the part of the infant to initiate an interaction, as in calling to the caregiver. Often in VP, infants appear to be manipulating or practicing parameters of which speech sounds are composed (pitch, duration, loudness, and spectral characteristics) without any social intention.

Coders are instructed not to treat protophones that are also interpreted to be "complaints" (whining) as instances of VP, since whining can be thought of as motivated by discomfort or frustration rather than exploratory interest, and it seems clear that some whining is socially directed. The first questionnaire item in Phase 2 specifically asks coders to designate how

many of the protophones in the segment were complaints, the illocutionary category indicating whining. This question is intended to supply an additional method of focusing coder attention on designating VP (the third question) only in cases where infants are not expressing distress, and indeed the coders of the present data seemed to abide by the instruction since they rarely rated segments (<0.5%) in such a way that the sum of the VP and complaint ratings was greater than 6—such a sum could be taken to mean the coder had assigned too much time in the segment to either VP or complaint.

*Turn-Taking* (TT) (Phase 2, question 4) is defined to refer to infant protophones produced during audible back and forth of vocalizations and speech between the infant and another person, respectively (i.e., caregiver, sibling, etc.). TT in this definition requires that the infant be perceived as responding to infant-directed speech, not merely vocalizing *during* caregiver talk, which is often directed to other speakers. Coders are encouraged to use an intuitive definition of the notion "conversation", that is, a period during which parties are perceived to produce and sometimes maintain a mutual exchange of vocalizations. The perceived end of a conversation often appears to correspond more or less to a five-second rule; that is, any period of five seconds or more during which no vocalizations occur is typically deemed to be a boundary between conversations, and the next occurring vocalization is treated as potentially starting a new conversation.

## Additional coding criteria for VP and TT

Cries and whimpers, which are also coded in Phase 1, are never treated as either VP or TT in Phase 2, since such sounds are not protophones at all. Similarly, laughter is not categorized as protophone material and so is not treated as either VP or TT.

The coders are instructed to ignore (i.e., not code at all and not include in judgments of VP or TT) any vocalization of either the infant or any other speaker unless the vocalization is salient in terms of amplitude and/or duration. In essence, utterances are not coded if their saliency is so low that the coder deems them to be not likely to have been noticed by the infant (as a listener to other-speaker utterances) or by other speakers (as listeners to the infant). Also, vegetative utterances (coughs, burps, etc.) are not coded, nor are effort grunts.

The goal of these procedures is to limit the judgments of VP to utterances deemed to be voluntary precursors to speech, not inspired by discomfort or desire to attract attention, and of sufficient saliency to be noticed by a nearby caregiver. Also, the saliency criterion limits counting of speaker utterances other than the infant to those that seem most likely to have potential effect on the infant.

VP and TT, in accord with our definitions, are in principle independent of each other—no infant utterance can properly be treated as both. Thus, VP is not socially directed and excludes socially interactive vocalization even if the interaction is deemed to be playful. Put another way, any protophone produced in vocal interaction is deemed TT, but not VP, even if the interaction is thought to be playful. In addition, some protophones produced in the absence of vocal interaction, especially those produced with low intensity, may be judged as pertaining to neither TT nor VP, at the coder's discretion.

Although TT and VP are in principle mutually exclusive, a particular segment can of course include both TT and VP at different points in time. We have previously discussed the potential for VP to function as a catalyst for parent-initiated social interaction [31]. If, for example, the infant is playing with a new sound, parents hearing the playful activity may repeat this sound to the infant, initiating a back-and-forth vocal sequence. Thus, in segments with high infant vocal activity containing both interactive and non-interactive protophones, a designation by the coder of Some TT *and* Some VP can occur. Thus, a rating of 3 for both would suggest that

about half the protophones in the 5-minute segment were produced in TT and that about half of the others were VP. The coders assigned ratings for VP+TT of > 6 on a particular segment in only 6.5% of the coded segments. 62% of the ratings for individual segments were < 6 for VP +TT, suggesting that coders judged many protophones to be neither VP nor TT. About 20% of segments were judged to include at least some complaint protophones, which were not supposed to be assigned to either VP or TT, and these segments seem to have contributed to the fact that far fewer than half the segments were given VP+TT ratings as high as 6. But there appear to have been additional cases where coders chose not to treat some protophones as either VP or TT, perhaps simply because they were not sufficiently confident in making the relevant judgment.

A recurring difficulty for coding occurs in cases where multiple speakers (sometimes as many as five or six in these recordings) are involved in a segment. Sometimes infants are silent during such periods, but on occasions where they do vocalize, it is often ambiguous whether they are trying to break into a conversation (bidding for attention, presumably engaging in neither VP nor TT), whether they are trying to respond to utterances that are not directed to them (attempting to engage in TT even without IDS), or whether they may simply be stimulated to vocalize in a playful way in the context of group vocal activity (which the coders could conceivably interpret as VP). Coders sometimes choose TT and in other cases VP to characterize such infant utterances. On yet other occasions they appear to treat the utterances in such segments as neither TT nor VP. The choice is left up to the coders as intuitive judges, in the same way that we presume caregivers make implicit judgments about infant motives.

## Syllable coding

In Phase 2, the 8 selected segments for each recording are coded in real-time for infant canonical and non-canonical syllables. A keystroke for each syllable provides the data. Listeners identified a total of 13,775 canonical syllables and 94,269 noncanonical syllables across the segments of the present study. To measure the emergence of advanced vocal forms, a *canonical babbling ratio* (CBR) was calculated as the total number of canonical syllables divided by the total number of syllables in each segment. Means and standard deviations of CBRs were calculated for each infant at each age. Three of the families did not complete a monthly recording at 7.5 months, and one did not complete a recording at 12 months. In cases where there were multiple recordings within an age for an infant, the means and SDs of these recordings were averaged prior to analysis.

## The intuitive coding principle

Trainees for coding in the OLL are encouraged to make judgments intuitively. We have long argued that normal human listeners must have been evolved not only to recognize speech and speech-like sounds, but also to be able to make reliable judgments about infant purposes in vocalization [32], for example, whether an infant is turn-taking with them, complaining, or instead comfortably exploring vocalization independent of interacting with anyone. As parents, human adults would be, we argue, at a substantial disadvantage in competition with other parents if they could not recognize such sounds and the functions infants serve with them, because they would not be able to adjust to infant needs and communicative capabilities. Our training procedures, then, attempt to help coders bring to conscious awareness their inherent capabilities to perceive infant sounds and to judge their purposes, providing a categorical terminology for describing the observed events that presumably fit well with naturally available human intuitions; see [31, 32] for additional theoretical perspectives on making intuitive judgments of infant vocal functions.

## Coding training and procedures

The six-to-eight-week coding-training procedure of the OLL is detailed in a recent publication from our laboratory [33]. For the present work, coders were blinded to all diagnostic and demographic information associated with each infant recording throughout the coding process. The training involved example presentation and supervised coding of segments from all-day recordings of infants from various demographic conditions, but none of the training segments were a part of the analyzed dataset for the present study.

Coders in the OLL are trained in phonetic transcription in their regular program of study in speech-language pathology. Usually, 6–10 trainees are initiated at the beginning of the first semester of a school year, with a two-hour introduction to infant vocal development using audio and audio-video examples presented primarily by the last author, a long-term investigator of human infancy. All other training personnel are required to have passed the same training and to have had considerable experience in coding infant vocalizations. Trainees are also introduced at the beginning to AACT [88], our coding software environment, facilitating a wide variety of coding and analysis tasks, and described in detail in the Supplementary Information to a prior publication [97]. Coders then engage in several weeks of practice coding in AACT with recorded 5-min segments of the same sort they will ultimately code for the research. They meet with the last author at least weekly in a group for supervised review of their coding. In addition, each coder meets individually with the last author for coding review. In order for trainees to be admitted to coder status, they are required to meet preset agreement standards for all the coding categories that are the focus of the present study. For details see Supporting Information to a prior publication [33].

## Coder agreement

Inter-rater agreement was examined for VP ratings, TT ratings, and CBRs in two ways: In Method 1 we drew coder agreement data from two rounds of coding on 105 five-minute segments extracted from LENA recordings of infants at the same ages as in the present study, coded by 7 of the same 15 graduate student coders who provided the analyzed data for the present study. The work followed the same coding protocol used in the present study and in prior efforts [33, 34, 97, 98]. Each of the 7 individuals had coded a subset of the 105 segments, and for the agreement effort, each was assigned to code a different subset assigned to a different coder originally. All these 15 have now graduated and are no longer available as coders.

In Method 2 we conducted a new, special re-coding of 60 five-minute segments from the present dataset of recordings, where all 15 original coders were represented, each having coded 4 segments of the 60, and where 4 new coders, still available in the project, re-coded all 60 segments, being blind to the original coding. The 60 selected segments were balanced to represent all three ages (20 from each). In addition, under Method 2, the four new individuals coded 60 segments that were randomly selected from the 21 Phase 1 segments for each of the recordings. Among those 60, fourteen had also been selected by the AACT algorithm for Phase 2 coding, and consequently the total number of segments coded by the four new individuals for the agreement effort was 106 rather than 120. Below we summarize the agreement outcomes, and for details see S4 Text.

*Method 1 outcomes*: Correlations between segment-level outcomes for the original coding by the 7 individuals and the blind second coding by the 7 individuals reassigned to work on different subsets of the 105 were weak to moderate in magnitude though highly statistically significant for VP, $r = .371$, $p < .001$ and TT, $r = .373$, $p < .001$. The correlation for CBR, was large $r = .872$, $p < .001$. The lower correlations for VP and TT are expected given the difficulty of recalling the events of any 5-min segment at the point of the questionnaires along with the

subtlety of making decisions about how to rate segments on the 5-point Likert scale. In contrast, the registration of canonical and non-canonical syllables is done in real-time while listening to the 5-minute segments with keystroke coding of each syllable and has proven to yield high agreement in past research as well [99]. Another way to assess agreement on VP and TT for the present purposes is to consider the level of agreement in terms of mean ratings. The VP ratings were only 8% higher in the re-coding than in the original, and the TT ratings were only 2% higher.

Moreover, because this agreement factor corresponds to the outcome data, we can conclude that the mean VP rating was more than 2.5 times higher than the mean TT rating in both the original and the agreement coding. Thus, the agreement data unambiguously suggest that if a different set of coders had supplied data on rates of VP and TT, the outcome conclusion regarding the rates of VP and TT would have been the same: VP occurred at a much higher rate than TT.

*Method 2*: Correlations for the 60 segments that were coded in their entirety by the four new coders with respect to the coding done by the original 15 coders (each of whom coded a subset of the 60) were similar to those found with Method 1. As with Method 1, the correlations (averaged across the four new coders with respect to the original 15 coders) were weak to moderate in magnitude though highly statistically significant for VP and TT (for VP, $r = .477$, $p < .001$ and TT, $r = .359$, $p < .005$). The correlation for CBR, was again large $r = .714$, $p < .001$. And again, the average VP and TT ratings were similar to the original ratings (VP 2% lower, TT 18% higher).

As in the case of Method 1, the agreement data conformed to the data from the original coders on the outcome. Both the new coding (averaged across the four individuals) and the original coding yielded much higher VP than TT values (new coders 2.3 times higher, original coders 2.9 times higher). Again, the agreement data unambiguously suggest that if a different set of coders had supplied data on rates of VP and TT, the outcome conclusion would have been the same: VP occurred at a much higher rate than TT.

For the 60 randomly selected segments, original coding was not available for Phase 2 (Phase 2 segments had been selected non-randomly by the AACT algorithm, see above), but the four new coders could be evaluated with regard to their agreement with respect to each other. Their correlations with each other across the six possible pairings of four coders were moderate and highly statistically significant for VP, $r = .57$, $p < .001$ and TT, $r = .45$, $p < .001$. The correlation for CBR, was again large $r = .77$, $p < .001$. The coders showed mean ratings that differed from each other by an average of 3% for VP and by an average of 20% for TT. And again, the difference between the mean VP ratings was substantially higher than the mean TT ratings for all four coders (VP on average 2.4 times higher). Thus, the agreement data on the randomly selected segments suggest that the higher rate of VP than TT was not a product of the non-random selection process for Phase 2 segments.

In all cases the correlations between agreement and original coders were highly statistically significant, and consequently all the statistical comparisons associated with the propositions were justified. The danger when coder agreement may be thought to be low is not Type I error, because coder disagreement introduces noise into the data, and consequently the danger is Type II error, failure to recognize real differences [100].

## Statistical approach

The data we present here are complex especially because they show an imbalance of amounts of VP and TT (VP far more common), and because they show a complex arrangement of segments that were categorized as having 1) both VP and TT, 2) neither VP nor TT, 3) some VP

but no TT, or 4) some TT but no VP. Consequently, interpretation of statistical analyses requires careful thought, and we have chosen to approach the matter eclectically. In each case, we have provided raw data with computed mean comparisons, effect sizes, and in some cases t-tests.

We used Generalized Estimating Equations (GEE) [101] for some of the more formal analyses involving Age as a factor. GEE is an advanced form of modeling that is well-suited to longitudinal data, because infants are not independent from themselves across ages. ANOVA as a method to evaluate longitudinal data violates the independence assumption. GEE is nonparametric and requires no independence nor normality assumption, accounts for correlations across variables and participants, and projects data in such a way as to correct conservatively for missing data. One of the advantages of using GEE analysis in the present case is that the imbalance in amounts of VP and TT as well as overlap of cells as seen in crosstabulation of VP and TT (see Table 1) do not adversely affect the analysis. The GEE method tends to yield fewer significant results than traditional approaches to analysis, so we have provided analyses by the conservative GEE approach where possible.

## Results

Proposition 1: The raw data showed a massive difference between the amounts of VP and TT. For our analyses in the main text, the Likert-scale ratings have been simplified: No VP was assigned a rating of 1, while any rating from 2–5 was treated as Some VP, with all Some VP

**Table 1. Distributional statistics for VP and TT.** The data revealed extremely different distributions of VP and TT for the 40 infants. All the data were calculated based on questionnaire responses of coders for the 1572 five-min segments. **A**. More than 90% of segments were judged to have some VP, while only 20% of segments were judged to have some TT. **B**. Crosstabulation showed that the vast majority of segments (74%) were judged to have Some VP and No TT. Only 1% of segments showed Some TT and No VP. **C**. The data breakdown in terms of the individual 1–5 ratings provided by coders for each segment showed that there were no segments judged to have TT occurring throughout the segment (a rating of 5), while VP was rated to have occurred throughout 31% of segments. Among the segments judged to have some TT, the vast majority were deemed to have occurred less than half the time (a rating of 2), while ratings suggested more than ¾ of the segments had VP occurring during at least half the time.

| 1A. Amount of VP and TT | | | 1B. Crosstabulation of VP and TT | | |
|---|---|---|---|---|---|
| **Number of segments** | | | **Number of segments** | | |
| **Amount** | **None** | **Some** | | **No VP** | **Some VP** |
| **VP** | 120 | 1452 | **No TT** | 99 | 1159 |
| **TT** | 1258 | 314 | **Some TT** | 21 | 293 |
| **Proportion of segments** | | | **Proportion of segments** | | |
| **Amount** | None | Some | | No VP | Some VP |
| **VP** | 0.08 | 0.92 | **No TT** | 0.06 | 0.74 |
| **TT** | 0.80 | 0.20 | **Some TT** | 0.01 | 0.19 |

| 1C. Amount of VP and TT by Likert Ratings (1–5) | | | | | | |
|---|---|---|---|---|---|---|
| | | **Number of segments** | | | | |
| | | **None** | **Some** | | | |
| | **Rating** | 1 | 2 | 3 | 4 | 5 |
| | **VP** | 120 | 212 | 318 | 440 | 482 |
| | **TT** | 1258 | 262 | 40 | 12 | 0 |
| | | **Proportion of segments** | | | | |
| | | **None** | **Some** | | | |
| | **Rating** | 1 | 2 | 3 | 4 | 5 |
| | **VP** | 0.08 | 0.13 | 0.20 | 0.28 | 0.31 |
| | **TT** | 0.8 | 0.17 | 0.03 | 0.01 | 0 |

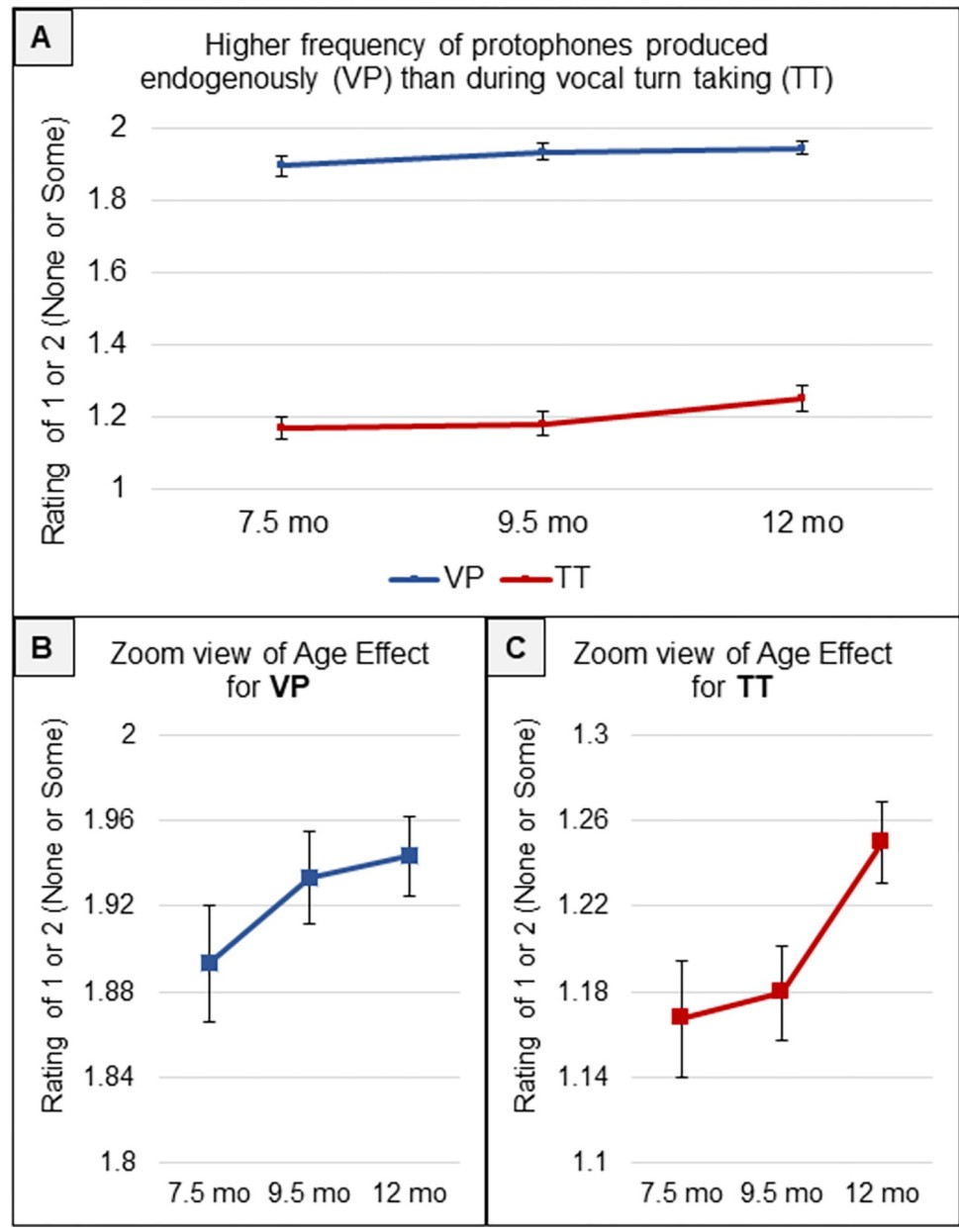

**Fig 1. Amounts of VP and TT. A.** The coders indicated that at all three Ages, the amount of VP for the 40 infants was much higher than the amount of TT. In the display, the scale has been converted so that 1 indicates No VP or TT in a coded segment, while 2 indicates Some VP or TT (any coded value from 2–5) occurred within a segment. On average, VP was deemed to have occurred considerably more than half the time during the segments, while TT was deemed to have occurred far less than half the time. The error bars are 95% confidence intervals. **B.** Zooming in so that mean differences and error bar information are more visible shows that VP occurred more often at 9.5 and 12 mo than at 7.5 mo. **C.** Zooming in also shows that TT occurred more often at 12 mo than at the younger Ages.

values converted to 2. The average rating for VP on this 1-to-2 scale was 1.92, indicating that far more than half the segments were judged to have Some VP. The average rating for TT, in contrast, scaled in the same way, was 1.20, indicating that far fewer than half the segments were judged to have Some TT. Fig 1A shows that this pattern of massive difference occurred in the raw data at all three Ages, and the 95% confidence intervals make clear that the differences

were highly significant (at all three Ages, $p < .0001$) with extremely large effect sizes (Cohen's $d$ = 2.7, 3.0 and 2.7 at the three Ages respectively). S5 Text displays the data on VP vs TT computed to reflect all 5 levels of the Likert scale ratings. The differences between amounts of VP and TT were so large that additional statistical comparison is superfluous.

However, Fig 1A suggests an effect of Age (something we did not anticipate), so we conducted GEE on Age as the independent variable and VP or TT as the dependent variables in two analyses. We found a highly significant Age effect ($p < .001$) for VP. Higher VP occurred at the older Ages as illustrated by the zoomed portrayal in Fig 1B. However, the effect was small ($d$ = 0.17). A similar GEE with TT as the dependent variable showed significantly higher TT ($p < .004$) especially at 12 mo, as shown in Fig 1C, but again the effect size was small ($d$ = 0.21). The plots of the estimated means and 95% confidence intervals based on GEE for these Age effects are provided in S5 Text.

Table 1 breaks down the data on the extreme differences in the distributions of VP and TT ratings. While more than 90% of segments were rated as having Some VP (ratings from 2–5), only 20% of segments were rated as having Some TT (Table 1A), and among the segments that were coded as having TT, more than 80% were rated as 2 (Table 1C), indicating extremely few segments were ever judged to include TT as much as half the time. In contrast more than three-quarters of segments were judged to include VP at least half the time, and more than 60% were judged to have VP more than half the time.

Crosstabulation analysis (Table 1B) shows that the great majority of segments had No TT while having Some VP, and it was extremely rare for segments to be judged as having Some TT and No VP (only 1% of segments). A breakdown of the distributional imbalances by Age is provided in S1 Table.

Proposition 2: These distributional imbalances highlight the importance of careful interpretation of analyses where CBR is the dependent variable in the cases where VP and TT are treated as levels of an independent variable, Vocal Activity Type. There was notable overlap of VP and TT occurrence within segments, as indicated in the Crosstabulations. Thus, our assessments of CBR by amount of Vocal Activity (VP or TT) are evaluated first separately in Propositions 2 and 3, and then together in Proposition 4.

The raw data displayed in Fig 2A show distinctly higher CBR for segments with Some VP (mean = .119) as opposed to those with No VP (mean = .027), and the pattern occurred at all three Ages. GEE showed a significant interaction of VP by Age ($p = .006$), reflecting the greater difference between CBRs for Some VP and No VP at 12 months than for Some VP and No VP at 7.5 or 9.5 months. The more conservative analysis by GEE also revealed significant main effects of Age ($p < .001$) and VP ($p < .001$). The GEE estimated means and CIs are presented in S5 Text.

The interpretation of the CBR difference requires taking into account the fact that 1,452 segments had Some VP, with only 120 having No VP. The mean difference for the raw data aggregated across Age, as in the GEE analysis, was highly significant by unpaired t-test ($p < .0001$, $df$ = 69), showing a small to medium effect size ($d$ = 0.36). Another perspective on the size of the mean difference between CBR for VP and TT (.092), however, is that it *exceeded* the well-known growth in CBR by Age—based on the present data, the mean difference from 7.5 to 12 months was .081.

Proposition 3: The outcome on CBR for TT was similar for VP. Segments with Some TT (314 segments) showed CBR higher (mean difference = .089) than segments with No TT (1258), as reflected in Fig 2B, although the difference was most notable at 9.5 mo. GEE revealed significant effects of Age ($p = .017$) and TT level ($p = .006$), with no significant interaction (see S5 Text for GEE estimates).

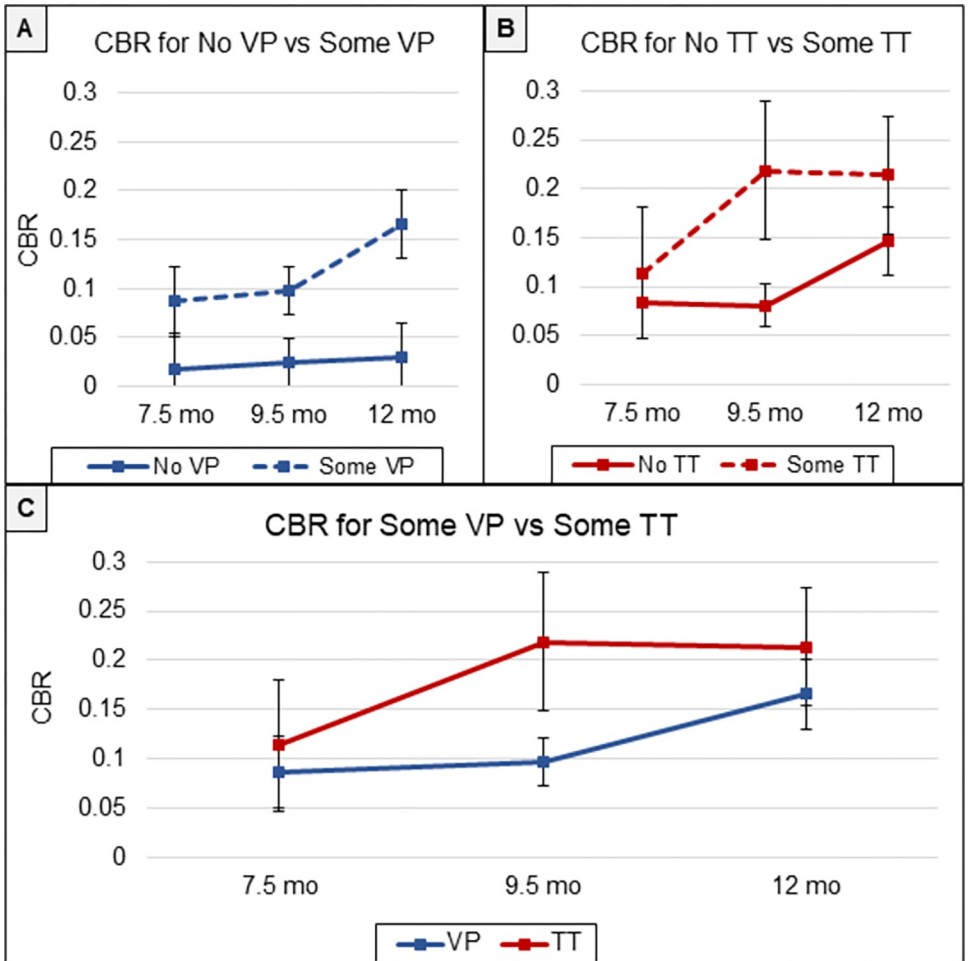

**Fig 2. Canonical babbling ratios as a function of VP and TT. A.** CBR was higher in segments judged to have Some VP than No VP. The difference was clear at all three Ages. **B.** CBR was higher in segments with Some TT than in segments with No TT. The pattern was most notable at 9.5 mo. **C.** Comparing the data for Some VP and Some TT, the data showed significantly higher CBRs for segments with Some TT than for segments with Some VP. The error bars for all the panels represent 95% confidence intervals.

Evaluation of the raw data aggregated across Ages by t-test showed the mean difference between No TT and Some TT (.089) was statistically significant ($p < .001$, $df = 75$), with a small effect size ($d = 0.271$). But as in the case of VP, the mean difference can be viewed as quite large from the perspective that it *exceeded* the range of average CBRs across the three Ages (.081). The raw data revealed that at 7.5 months, the infants showed a small change in CBR from No TT to Some TT with a relatively low CBR overall ($d = 0.08$, mean difference = .031). The largest CBR difference between No TT and Some TT was observed at 9.5 months ($d = 0.38$, mean difference = .138).

Proposition 4: Fig 2C combines the raw data for CBR to compare segments with Some VP and Some TT. As predicted, CBR was significantly higher at Some TT. GEE (see S5 Text) revealed a significant effect of Vocal Activity Type (VP vs. TT, $p = .021$), with no interaction and an Age effect with marginal significance ($p = .055$).

Aggregating the raw data across Ages, the mean difference between Some TT and Some VP was significant by t-test ($p < .005$, $df = 75$). The effect size was small ($d = 0.23$), but again the mean difference (.075) was not small with respect to the range of CBRs found across ages,

representing more than 90% of that range. And again, the difference was most pronounced at 9.5 months ($d$ = 0.37, mean difference = .122).

Not directly displayed in Fig 2 is the fact that that there was a substantially higher mean CBR at No TT (.105) than at No VP (.027). This difference may primarily reflect the fact that 73% of the 1,258 segments with No TT were included within the 1,452 segments with Some VP. The similar levels of CBR for No TT and Some VP (.119), appear to reflect that overlap— the majority of all segments in the sample included No TT but Some VP. The difference between CBR at No TT with respect to No VP may also be related to a greater amount of crying/whimpering and whining in segments with No VP (see S5 Text). The higher CBR for segments with Some TT than Some VP should also be considered in terms of overlapping categorizations: 273 of the 314 segments coded as having Some TT were among the 1,452 that were coded as having Some VP.

## Discussion

Our results support the supposition that human infants are motivated to vocalize on their own, to produce speech-like sounds (protophones) in the first months of life copiously, most frequently with no obvious intent for anyone to hear them. The coders deemed infants to have engaged in at least some exploratory vocal play, that is to say, they were deemed to have produced protophones without social directivity, in more than 90% of the five-minute segments evaluated. In contrast, the use of protophones in vocal turn taking with caregivers occurred far less frequently, with just 20% of five-minute segments being judged to include any vocal turn taking at all. Perhaps more importantly, exploratory vocal play was judged to have occurred more than half the time during the majority of coded segments, while vocal turn taking was judged to have occurred as much as half the time in only 4% of the segments.

These results present a fundamental challenge to the most widespread view of infant vocal development, wherein it has been expected that infants learn to produce speech-like sounds primarily through engaging in vocal interaction and by imitating their caregivers. In fact, the results suggest the possibility that infants learn to produce speech-like sounds mostly by exploring their own vocal capabilities repeatedly, every day, and thus learning about and elaborating the categories of sound they can produce. We cannot, based on current evidence, rule out the possibility that most of the progress in speech-like vocalization of infants that occurs across the first year is the product of this exploratory activity.

The importance of extensive, socially unengaged vocal activity makes sense if we think of infants as agents of their own development rather than as responders to caregiver entreaties. Of course infants must ultimately learn many aspects of phonology from caregivers, but caregivers may not be able, even if they wish to, to spend much of their time attending directly to infants, and they may not be able to control the nature of infant attention very effectively even when they have the time to try [17]. As a result, it is not clear how much is learned by infants about phonology from exploration and how much from listening and attempting to mimic, especially in the first year, where the great majority of that learning might be the product of endogenously generated exploration.

The fitness-signaling hypothesis [52, 54] offers perspective on how the human system of vocal development emerges to involve so much endogenous activity and to be so different from vocal patterns in other primates, even within the first year of life. The idea that hominin infants have been under selection pressure (due to their relative altriciality) to produce wellness indicators can help explain the apparent deep-seated motivation of humans to engage in playful, exploratory vocalization, while other primates (being less altricial) do so rarely if at all. Presumably, human caregivers notice infant vocal play—even sometimes when it is not

directed to them—as a sign that an infant is well if the vocalizations display comfort and/or vocal competency. Even whiny vocalizations (and cry for that matter) can supply useful information about wellness to caregivers if these expressions of negativity are produced effectively and not too often.

Infant protophone production is thus thought, in accord with the fitness-signaling hypothesis, to have provided a rational basis (in addition to other fitness signals, such as gross motor control, feeding adequacy, resistance to infection, etc.) throughout hominin history for caregivers to allocate their investments toward infants who show particularly high potential for surviving and providing them with grandchildren. Parents and other caregivers were thus, in accord with the hypothesis, the agents of the selection pressure on protophone production. Hominin infants were accordingly selected to not only be able to produce speech-like sounds, but to desire to do so, to find it rewarding to explore the vocal space in much the same way that all primates seem to find it rewarding to explore the physical space, for example with their hands. With such inherent desire, infants produce huge numbers of protophones, even when there appears to be nobody listening.

These interpretations fit well within an evolutionary developmental biology (evo-devo) framework [102–105]. Evo-devo encourages recognition of the tendency for evolutionary change to occur through changes in developmental patterns and schedules. Garstang [106] argued a century ago that significant changes in organisms occur preferentially by natural selection for changes in infancy that can form foundations for additional changes across the lifespan. The idea has been extensively supported with evolutionary biological research in more recent time [104, 107, 108]. The fitness-signaling hypothesis for human infant vocalization is embedded within this framework. It suggests that human infants develop vocally by virtue of having been selected to explore vocalization, because that kind of exploration has resulted in systematically greater caregiver investment. Secondarily, that exploration laid foundations for later developments that would ultimately yield language, and the advantages of language may have yielded additional selective pressure on human infants to explore vocalization and vocal interaction. We have argued that without the developments associated with both exploratory vocal play and active vocal turn taking, none of the subsequent developments required for language could have occurred [54, 55].

It seems no surprise, if the hypothesis is on target, that elaborate and thus particularly informative protophones, the ones including canonical syllables, should be preferentially produced during periods of notable vocal play. Indeed, when infants were engaged in exploratory vocal play, their protophones were more advanced, manifesting higher amounts of canonical babbling than when infants were not engaged in vocal play. Similarly, when infants were engaged in vocal turn taking, their protophones showed more canonical syllables than when they were not so engaged. These facts would appear to support the fitness-signaling hypothesis because the circumstances of high vocal activity and of vocal interaction are the very circumstances where caregivers are most likely to pay conscious attention to the infant's capabilities and inclinations.

Perhaps the very most revealing circumstance that we were able to monitor to assess the production of infant vocal fitness signals was during the relatively uncommon circumstance of vocal turn taking. The very highest levels of canonical babbling occurred during vocal turn taking, with canonical babbling ratios (CBRs) > .20 at both the 9.5- and 12-month ages. Prior research based on laboratory recordings has designated a nominal value of .15 CBR as indicating an infant to be in the canonical stage [89, 109, 110]. The literature shows that laboratory-based recordings, where parents typically attempt to elicit vocalization from infants, have tended to yield higher CBRs than we observed in these all-day recordings. The discrepancy makes sense if we recognize that the vast majority of the family activity in the samples from the

all-day recordings did not include vocal interaction with the infants. Even the samples with some vocal turn taking appear not to have included as much turn taking as in typical laboratory recordings.

There can be no doubt that humans are highly social. Clearly, early hominins' relatively large living groups necessitated a high level of social bonding, which created a need for an efficient communication method, resulting in positive selection pressures on the evolution of vocal communication [111–113]. Chevallier and colleagues [114] have noted that "social motivation constitutes an evolutionary adaptation geared to enhance the individual's fitness in collaborative environments" (p. 2). So there is no denying that the human infant must have been selected to be interested in other people and to begin the process of adapting to human sociality from very early. And yet the overwhelming vocal activity type seen in the present work appeared to be endogenously generated.

Canonical babbling is well-established as a robust stage of development, known to emerge even in infants at risk because of premature birth or very low socio-economic status [115, 116]. Both in typically-developing infants and in infants with or at risk for various disorders [94, 99, 117–120], canonical babbling emerges after the middle of the first year, we propose, as a self-organized product of vocal exploration in the earlier months of life, and from that point forward, infants are in a position to produce at least some word-like vocalizations that can be clearly identified as such by parents [30, 72, 86]. The current results on the prominence of canonical babbling in the very circumstances of potentially most social benefit to the infant, during vocal play and vocal turn taking, suggest infants adapt by providing the highest quality vocal indicators at those times where caregivers may be most attentive. In addition, canonical babbling in those circumstances presumably provides caregivers with a maximal opportunity, not only to assess infant wellness, but to provide contingent feedback, a potentially critical scaffold for conversational learning and later language development [10, 84, 85].

A finding we did not predict, although perhaps we should have, was that both TT and VP occurred at somewhat higher rates at the older ages. It makes sense to suggest that infants ramp up their vocal activity in all circumstances as time passes, as the possibility of genuine linguistic communication approaches, and as the priority on learning both through exploration and through interaction grows.

While fitness-signaling explanations for endogenous vocalization, social motivation to vocalize, and the tendency to produce high rates of canonical babbling during VP and TT respond to Tinbergen's *ultimate* questions (i.e., about evolution and survival value), they do not respond to Tinbergen's *proximal* questions (i.e., about development and causation). The proximal questions have been and are being addressed by a larger literature on development, including experimental studies [17, 121, 122], longitudinal observational work [13, 73, 123], and significant theory building. Vihman's articulatory filter proposal [124] has achieved particular prominence among portrayals of the proximal reasons for infant vocalization patterns and their role in vocabulary acquisition. This literature can be viewed as supplying a backdrop for interpretation of much of the reasoning presented here regarding the evolution of human vocal tendencies.

## Measurement limitations

The coding approach we used is founded in the notion that human judgments must be the gold standard for any measure of vocal development, because human caregivers impose the ultimate naturally selecting constraints on the infant vocal system. In any case, there is no automated method to date to replace human coding for the measures at stake here. Of course, one might imagine a more reliable approach to human coding for VP and TT as well as for

CBR, an approach that might require repeat-observation coding of each infant utterance in context. But time constraints on the coders, who were students in a rigorous program, prohibited us from using repeat-observation because it requires at least tenfold more time to code the same materials. See our prior work [31] for an analysis using the more time-consuming method on a much smaller dataset.

Thus, to measure social and exploratory vocal functions, coders were required to use an intuitive and practical method, estimating on a Likert scale how often infants engaged in VP and TT for each segment. This measure, obtained immediately after real-time coding for CBR on each segment, is clearly a blunt instrument, subject to only fair point-to-point inter-observer agreement. Yet the agreement was considerably and highly significantly better than chance. More important than the absolute level, however, agreement on the key issue of Proposition 1 (the much higher estimated VP than TT) was very high indeed, with every coder in every test showing considerably higher VP than TT, always at $p < .001$. Coder agreement can be problematical when differences between targeted comparisons of variables are small with respect to coder differences—one runs the risk of Type II error, because low coder agreement can impose a power limitation, acting as noise in the evaluation and making it less likely to discern real differences. But the data suggest there was no such problem for the VP vs TT comparison in the present data because the differences were so large that coder disagreements could not have obscured them.

A limitation of this study is that canonical babbling was the only measure we used to assess advanced vocal forms. Future studies could, with sufficient time and money for much more elaborate coding or more advanced automated analysis, compare variation in infant vocal types based on phonatory characteristics across contexts, infants' recombination of syllables at the utterance level, and more specific contextual social or environmental information such as response contingency and infant- and adult-register in infant-directed speech.

## Interpretive issues for CBR comparisons

The present research did not acquire data from an experiment where one might insist on two exclusive conditions where all utterances by infants would be either TT in one case or VP in the other. These were instead observational results from natural home activities where both TT and VP occurred frequently within the same 5-minute periods. Our CBR comparisons are complicated by the differing distributions of VP and TT, along with the unavoidable overlap of segments having some of both VP and TT. A major advantage of using GEE is that this statistical method is not adversely affected by imbalances in cells nor by overlaps as seen in crosstabulation—GEE analysis offers assurance that statistical differences produced by the method are real.

Yet it remains worthwhile to take note of the fact that the difference between CBRs at No VP and No TT were clearly affected by the distributional imbalance; only about 8% of segments had No VP, while 80% had No TT, and notably, among those with No TT, more than 90% had Some VP. Consequently, the higher CBR for No TT than No VP may have been due largely to the presumable enhancing effect of VP on CBR in the No TT segments. Indeed, the similarity of the CBRs for No TT (mean = .105) and Some VP (mean = .119) appears to reflect the fact that the No TT segments were largely the very same ones that showed Some VP. It was rare indeed for TT to occur in the absence of VP (only 1% of segments), and segments with neither VP nor TT were also very uncommon (6%). The latter circumstance yielded extremely low CBR (.03 computed over the 99 segments with neither), an outcome that we interpret (cautiously, given the low number of segments) as reflecting the fact that there was no enhancing effect of either TT or VP on those segments. By the same token, the 293 segments with both TT and VP

showed much higher CBRs (> .16) than segments showing Some TT but No VP or Some VP but No TT, circumstances that yielded intermediate CBRs (.08 and .10 respectively).

Another possible contributor to low CBR for segments with No VP concerns vocal negativity. Segments with No VP showed rates of crying and whimpering (~ 3 such utterances per min) that were more than 5 times higher than segments with Some VP. Of course, crying and whimpering were not included directly in our analyses, but the state of infants during periods of such negativity may have affected CBRs and may have influenced coders to make the judgment that VP was low or absent. Similarly segments with No VP showed high rates of whining, as indicated by the questionnaire item on complaint, where the rating for segments with No VP was ~ 4.4 (complaining the great majority of the time) while the rating for segments with Some VP was ~ 1.8 (complaining less than half the time). In accord with the instructions to coders, no utterance should have been judged to be a complaint and at the same time an exemplar of VP. So, just as the negativity indicated by crying/whimpering may have influenced low CBR and ratings of No VP, it seems likely that segments with high complaint ratings would have caused both low CBR and ratings of No VP. Segments with No VP routinely showed high complaint ratings; in fact, 71% of segments with No VP were rated as having complaint throughout the segment (a rating of 5). Thus, it should be no surprise that when babies were crying or whining, they were not only not engaging in VP, but they were not producing very speech-like protophones. In contrast, segments with No TT showed low rates of both crying and complaint, and this pattern may have contributed to the higher CBR at No TT than at No VP. More data on these issues can be found in S5 Text.

## Conclusions

Our research suggests human infants are independent agents of their vocal development, exploring speech-like sounds (protophones) extensively throughout the first year, presumably practicing, calibrating, and elaborating their vocal capacities. These independent vocal activities appear both to inform parents about infant well-being and to establish critical foundations for later developments leading to language. In addition, infants in our studies appear inclined to engage in vocal interaction, during which they presumably supply caregivers with information about their developmental levels and their wellness, while also learning to converse and establishing additional foundations for language.

The research suggests that the vast majority of infant protophones across the first year of life are endogenously generated. During periods of exploratory vocal play, infants appear to produce especially high levels of canonical babbling, an advanced type of protophone. But the most frequent occurrence of canonical babbling appears to correspond to the much more rare occasions of vocal turn taking.

The results are consistent with the fitness-signaling hypothesis, which suggests the infant tendency to produce extensive speech-like vocalization has been naturally selected to signal wellness and viability to caregivers. Presumably, ancient hominin parents invested preferentially in infants whose protophones were most informative of fitness. The present data are also consistent with the idea that infants to this day provide wellness information to caregivers through protophone production and that they adapt the production of protophones to ensure that the most advanced of them occur at points in time when caregivers are most likely to notice them, during periods of vocal play, and especially during vocal turn taking.

## Supporting information

**S1 Text. On exclusion of sex and SES in the main analyses.**
(PDF)

**S2 Text. Abbreviated OLL coding instructions.**
(PDF)

**S3 Text. Questionnaire items for Phase 1 and Phase 2 coding.**
(PDF)

**S4 Text. Additional coder agreement data.**
(PDF)

**S5 Text. Additional data.**
(PDF)

**S6 Text. Algorithm for Phase 2 segment selection.**
(PDF)

**S1 Table. Distributions of VP and TT by age for all 5 Likert scale levels.** Amount of VP and TT by Likert-scale ratings by age.
(PDF)

## Acknowledgments

The authors wish to thank the participating families in Atlanta, GA, and the graduate student coders of the Origin of Language Laboratories.

## Author Contributions

**Conceptualization:** Helen L. Long, Ulrike Griebel, Megan M. Burkhardt-Reed, D. Kimbrough Oller.

**Data curation:** Gordon Ramsay, Edina R. Bene, Dale D. Bowman, D. Kimbrough Oller.

**Formal analysis:** Helen L. Long, Gordon Ramsay, Dale D. Bowman, D. Kimbrough Oller.

**Funding acquisition:** D. Kimbrough Oller.

**Investigation:** Helen L. Long, Gordon Ramsay, Ulrike Griebel, Edina R. Bene, Megan M. Burkhardt-Reed, D. Kimbrough Oller.

**Methodology:** Helen L. Long, Ulrike Griebel, Edina R. Bene, D. Kimbrough Oller.

**Project administration:** Helen L. Long, Gordon Ramsay, Edina R. Bene, D. Kimbrough Oller.

**Resources:** Helen L. Long, Gordon Ramsay, Ulrike Griebel, Edina R. Bene, Megan M. Burkhardt-Reed, D. Kimbrough Oller.

**Software:** Helen L. Long, Edina R. Bene, D. Kimbrough Oller.

**Supervision:** Gordon Ramsay, D. Kimbrough Oller.

**Validation:** Helen L. Long, D. Kimbrough Oller.

**Visualization:** Helen L. Long, Megan M. Burkhardt-Reed, D. Kimbrough Oller.

**Writing – original draft:** Helen L. Long, Gordon Ramsay, Ulrike Griebel, Dale D. Bowman, Megan M. Burkhardt-Reed, D. Kimbrough Oller.

**Writing – review & editing:** Helen L. Long, Gordon Ramsay, Ulrike Griebel, Edina R. Bene, Dale D. Bowman, Megan M. Burkhardt-Reed, D. Kimbrough Oller.

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
