## [Decision Letter · Decision Letter 0]

29 Mar 2022

PONE-D-22-03801Canonical babbling during vocal turn taking and independent vocal playPLOS ONE

Dear Dr. Long,

Thank you for submitting your manuscript to PLOS ONE. After careful consideration, we feel that it has merit but does not fully meet PLOS ONE’s publication criteria as it currently stands. Therefore, we invite you to submit a revised version of the manuscript that addresses the points raised during the review process.

I agree with Reviewer 2 that the abstract does not mention the two populations (children with LR vs HR for autism) and so seems to suggest a focus mainly on the drivers of canonical babbling. Please revise the abstract according to what the main emphasis of the study is, taking into account the comments below.

Both reviewers raise the issue of whether the ‘high-risk for autism group’ can be considered as low in social motivation so this should be addressed from the outset as it impacts on the formulation of your research questions.

I also agree with Reviewer 2 that the information about model structure and results should be included into the main text as this is what is being interpreted in the Discussion. In my own view, this is especially important since readers need to understand the motivation behind your choice of GEEs as opposed to LMEMs, whether and how this affects your method of contrast coding and its implications for whether effects are interpreted as simple vs. main effects. Furthermore, you mention several different models but do not seem to provide model comparison data. From your analysis code on the OSF I can see that some models coded age numerically and some used treatment coding of age groups, but it is not clear which models were selected for reporting and why. Overall, the section called ‘Statistical Approach’ needs to be significantly expanded to include revised parts of what is now Supplementary Information and more clarity with respect to your analysis strategy. This seems especially important as some models did show an effect of TT and some did not.

This may be related to the lack of clarity about your segment coding for Turn Taking vs Vocal Play and its treatment in the statistical models. I could grasp the distribution of TT vs VP across sessions only after consulting the cross-tabulation in the Supplementary Materials so to avoid confusion by the readers I strongly suggest moving the first two tables from Appendix A into the main text and remove Table 3 which strikes me as unhelpful as it obscures the relationship between TT and VP. It may make sense to streamline the cross-tabulation based on your dichotomization of the TT and VP ratings resulting in four types of segments: 1) lowTT/lowVP, 2) lowTT/highVP, 3) highTT/lowVP, and 4) highTT/highVP. Even though the highTT/highVP combination does not occur the lowTT/lowVP combination does and so I doubt that it is justified to analyse the data separately for TT and VP rather than run a model that includes both fixed effects. Your r-code on the OSF suggests that you did run such a model so please motivate which models you present in the paper and why or present a multiverse analysis.

Both reviewers make further  helpful comments on how to interpret findings with respect to the HR and LR groups for autism which I urge you to take into consideration. Unfortunately, a 3^rd^ reviewer, who is an expert in early infant vocalizations rather than in autism, was unable to provide a review. Still, I feel the paper needs a clearer theoretical interpretation for what your data show with respect to the role of social vs endogenous motivations for canonical babbling and how these two drivers may interact. You state in line 390 that the effect was larger for VP but you don’t seem to provide statistical evidence to back this up. Given the multiple statistical models you have run it is not clear what would constitute such evidence so I urge to provide a clearer link between you statistical modelling and your theoretical conclusions with respect to the motivations that may drive canonical babbling. In light of the reviewers’ comments on whether the HR group can be assumed to have lower social motivation or not you may want to de-emphasise the interpretation of the null effects of risk group in favour of a clearer theoretical discussion of the two types of babbling motivation.

Both reviewers have made many other helpful comments that I hope you will consider in your revision.

One minor point: For the first two models reported in the Appendix, you base your predictions on the social motivation theory which seems to be wrong with respect to the model containing VP as a predictor as this clearly pertains to the endogenous/play theory (I suspect this is a copying error in line 227 of the SI).

We look forward to receiving your revised manuscript.

Kind regards,

Vera Kempe

Academic Editor

PLOS ONE

Journal Requirements:

3. Please upload a copy of Figure 2, to which you refer in your text on page 14. If the figure is no longer to be included as part of the submission please remove all reference to it within the text.

Reviewers' comments:

Reviewer's Responses to Questions

**Comments to the Author**

1. Is the manuscript technically sound, and do the data support the conclusions?

Reviewer #1: Partly

Reviewer #2: Partly

2. Has the statistical analysis been performed appropriately and rigorously? 

Reviewer #1: Yes

Reviewer #2: Yes

3. Have the authors made all data underlying the findings in their manuscript fully available?

Reviewer #1: Yes

Reviewer #2: Yes

4. Is the manuscript presented in an intelligible fashion and written in standard English?

Reviewer #1: Yes

Reviewer #2: Yes

5. Review Comments to the Author

Reviewer #1: The article is very interesting and well written. It compares canonical babbling in infants at high or low risk for ASD in different socio-communicative situations. The main issue is that the conclusions driven from the results are limited to high vs. low risk, but the authors did not discuss that a certain - most likely high - proportion of infants at high risk for ASD will not be diagnosed with ASD later in their lives. This leads me to the following question: Why should high risk children who develop typically differ from low risk children in canonical babbling? Do you know how many of the high risk infants were later diagnosed with ASD? I do understand if the outcome of the children is not known to date, but it should be discussed as a limitation that it is unkown how many of the high risk children actually have ASD and how many of the low risk children actually develop typically. Please especially consider to rewrite the sentence on page 18, lines 379-381, and the first paragrah on page 20, taking into account the limitation mentioned above. Did you consider any inclusion criteria concerning pregnancy/delivery, i.e., preterm/term birth, potential complications, birth weight and size, head circumference, APGAR score, etc.? Are there group differences concerning these aspects?

Minor issues:

Page 3, line 54: Would it be more accurate to use wellbeing instead of wellness?

Page 10, line 196: Please rather use "from the first study month". Without the word "study" one may misunderstand it as the first month of life.

Page 11, lines 212-214: Please clarify the procedure of selecting 8 out of 21 segments. Did you take care for age-matching?

Page 13, lines 271-278: Please clarify: How can a segment contain more than half the time VP AND TT? In this case, VP and TT would need to occur simultaneously. Is this possible?

Reviewer #2: Thank you for the opportunity to review this manuscript, which presents interesting canonical babbling data and interprets it through an evolutionary developmental biology lens. This is a good study with strong methods and interesting results, but in the current format, some of the major claims in the discussion are unsupported by data presented in the manuscript. Specifically, significant space is dedicated in the discussion to interpreting the implications of the findings on the role of social motivation in babbling development, which is substantiated by null interactions that are presented only in the supplementary material. I think this can be largely resolved by moving these analyses to the main text. Based on the discussion, interactions with risk appear to be a main focus of the paper, and I think the paper would be strengthened if the introduction, methods, and results were edited to reflect that. Major and minor comments follow.

Major Comments:

Abstract

The fact that the sample includes infants at high and low risk for autism should be mentioned in the abstract.

Introduction

Why is it taken as a given that babbling is a signal of fitness rather than simply selected for because it is a good way develop language, which is itself beneficial?

If an entire section of the discussion will be dedicated to the lack of risk*TT interaction, this should be listed as a hypothesis. The risk*VP interaction hypothesis should probably be listed, as well.

Methods:

Regarding the low-high median split of SES: what is the point of this? It doesn’t seem like these groups are actually used for anything in the analysis, and as they are, they do not communicate much about the SES levels of the groups. It would be more helpful to anchor where that split fell in terms of the report of maternal education, or perhaps just report (maybe in the supplement) the maternal education levels reported by group.

I’m confused about the segment selection. First: “From these 21, eight segments, all having been coded as showing high volubility (infant protophone rate) and a range from high to low infant-directed speech were selected from each recording, yielding a total of 3799 segments.” I think this sentence is missing a word. Is this saying that from the 21 randomly chosen segments, the eight segments with the highest volubility for each participant was selected? If it was not based on the highest volubility, it would be good to clarify how the 8 segments were picked.

Second, I did not understand until I read the results that if an infant had multiple days of recordings that fell within a single “timepoint,” all of the recordings were used, meaning that infants had varying numbers of segments within a timepoint, and some infants contributed data ~1 month apart to the same timepoint. It would help the reader to make this more explicit in the methods.

Regarding the statistical approach, I don’t understand why the information about the GEEs was relegated to the supplement, but it should absolutely be in the main text. It was challenging to figure out what the analyses actually consisted of from the main text. Moving those two paragraphs to the main text would be good.

Results

Include the full interaction results, since these are heavily interpreted in the discussion.

Discussion

Social motivation section: It seems like a misreading of the social motivation hypothesis of autism to assume that the high risk group as a whole should be considered to have reduced social motivation. While it is reasonable to hypothesize that reduced social motivation is an endophenotype of autism, with an intermediate phenotype observed in those high risk siblings who do not ultimately receive an ASD diagnosis, it could alternatively be predicted that higher levels of social motivation among HR-no-autism-outcome siblings is precisely what protects them from an eventual ASD diagnosis. Thus, the HR group overall is not necessarily expected to have lower social motivation, even if the HR-ASD-outcome infants do.

Line 394 “The findings for CBRs during TT (alongside no major differences between risk groups) suggest similar levels of social motivation in both groups, with all infants showing the tendency to produce higher CBRs during segments rated as having any TT compared to those rated as TT.” These findings are not presented in the main text. Include the interaction effects to make this claim.

Also, there is no discussion of the fact that the authors examined the HR group as a whole, where there may be a more clear difference if they separately examined the HR infants who do and do not go on to receive an ASD diagnosis. For example, line 417: “Our results suggest that the most advanced prelinguistic vocal forms (i.e., canonical syllables) produced during early face-to-face interactions may be robust even in the face of a potentially elevated likelihood for social communication deficits.” I don’t really understand what this is saying. The HR group isn’t potentially elevated in risk, they are elevated in risk. But within that group are clear subgroups (that the authors may have access to) of infants who do or do not go on to actually have social communication deficits.

This paragraph/sentence can’t stand on its own (line 458) Interestingly, a preliminary analysis on VP data revealed a significant difference in CBRs between Risk groups over time, but only during periods of high VP (p = .039, b = .06, see SI Appendix C for additional analyses). Either mention the results of these preliminary analyses in the results and expand on this further, or don’t. As it stands, this sentence doesn’t even give the direction of effects.

Conclusion:

“Our results on the risk groups offer support for a potentially robust social motivation in infancy to produce higher rates of canonical syllables during interaction, even in the presence of an elevated likelihood for social communication deficits.” Again, this sentence is not substantiated by results presented in the paper as it stands, but would be fine if the null interaction is included.

Minor Comments

Intro:

“And it is especially important to determine the role of canonical babbling, because it is so much like mature speech, during these different circumstances.” This sentence could be strengthened to more clearly articulate the aim of the study.

This sentence was really hard to parse (intro line 118) “Our assumption is that both social and exploratory motivations are involved in the production of protophones—even during social interaction, one can imagine that playful exploration of sounds and perhaps relations of the protophones with the sounds of the interacting other speakers may motivate infant activity.”

Methods:

Missing a word (line 256) “For example, a TT rating of 5 was applied to segments where a caregiver was clearly speaking to the infant, and the infant was protophones in an apparent back and forth vocal interaction for essentially the whole segment.”

Results:

Line 324 “indicating an increased production of canonical babbling during periods with high social vocal activity compared to low social vocal activity (p < .001, b = .04).” This should say “any social vocal activity compared to no social vocal activity” to accurately reflect the split of data that was examined.

Discussion:

Line 380 “may be more resistant to cognitive effects of risk than has been expected.” I don’t understand what is meant by “cognitive effects of risk”

6. PLOS authors have the option to publish the peer review history of their article (what does this mean?). If published, this will include your full peer review and any attached files.

Reviewer #1: No

Reviewer #2: No

---

## [Author Response · Author response to Decision Letter 0]

27 Jul 2022

Please see the Response to Reviewers document for point-by-point responses to individual comments. We are grateful for the extension for making the necessary changes to this manuscript.

---

## [Decision Letter · Decision Letter 1]

21 Sep 2022

PONE-D-22-03801R1Infants vocalize most during independent vocal play but produce their most speech-like vocalizations during vocal turn takingPLOS ONE

Dear Dr. Long,

Thank you for submitting your manuscript to PLOS ONE. After careful consideration, we feel that it has merit but does not fully meet PLOS ONE’s publication criteria as it currently stands. Therefore, we invite you to submit a revised version of the manuscript that addresses the points raised during the review process.

Both reviewers, one of which had reviewed the original submission, highlight the improvement in response to previous comments, in terms of focus and methodological clarity. I agree that your submission makes an important contribution to the field and that the submission has done much to make this clear. Still, both reviewers feel that there are still outstanding issues in terms of theoretical scope as well as clarity about some of the methodological choices and features.  I therefore hope you will be able to undertake a further revision to address these concerns. I do not anticipate this to require substantial changes but rather clarifications, justifications and some additions to the Abstract and the Introduction as indicated in the Reviewers' comments.                                                                

We look forward to receiving your revised manuscript.

Kind regards,

Vera Kempe

Academic Editor

PLOS ONE

Journal Requirements:

Reviewers' comments:

Reviewer's Responses to Questions

**Comments to the Author**

1. If the authors have adequately addressed your comments raised in a previous round of review and you feel that this manuscript is now acceptable for publication, you may indicate that here to bypass the “Comments to the Author” section, enter your conflict of interest statement in the “Confidential to Editor” section, and submit your "Accept" recommendation.

Reviewer #2: (No Response)

Reviewer #3: (No Response)

2. Is the manuscript technically sound, and do the data support the conclusions?

Reviewer #2: Partly

Reviewer #3: Yes

3. Has the statistical analysis been performed appropriately and rigorously? 

Reviewer #2: Yes

Reviewer #3: I Don't Know

4. Have the authors made all data underlying the findings in their manuscript fully available?

Reviewer #2: Yes

Reviewer #3: Yes

5. Is the manuscript presented in an intelligible fashion and written in standard English?

Reviewer #2: Yes

Reviewer #3: Yes

6. Review Comments to the Author

Reviewer #2: Thank you for the opportunity to review this substantially revised manuscript. By shifting the focus away from evaluating the social motivation hypothesis and eliminating the infants with elevated likelihood for autism, the authors have addressed many of my initial concerns (and I look forward to their future paper examining these relationships in autism). The new manuscript is much more streamlined and provides significantly clearer rationale for the tests conducted and conclusions drawn, with a goal of evaluating the role of endogenous vocal activity in vocal learning. My main concerns at this stage are 1) that the selection of high volubility segments may influence the main conclusions drawn, and 2) that the low coder agreement on the Vocal Play (VP) and Turn Taking (TT) measures may significantly impact conclusions drawn about canonical babbling ratios (CBR) during these periods in ways that are not acknowledged. These concerns are explained in more detail below, along with some more minor suggestions.

Abstract

It would really help to spell out that VP and TT are neither mutually exclusive in segments nor encompass all kinds of vocalizations. At the least, changing “Periods of vocal play and turn-taking corresponded to elevated levels of the most advanced protophones (canonical babbling),” to “Periods of vocal play and turn-taking corresponded to elevated levels of the most advanced protophones (canonical babbling) relative to periods without vocal play or without turn-taking,” would help.

Testable Propositions

Why use “proposition” instead of “hypothesis?”

Proposition 2 – At this point, I did not understand how can a segment could have no VP without simply being TT. This is explained in great length later in the methods section, but should be briefly mentioned here (e.g., a brief mention of things like complaints).

Participants

I still feel that some direct information about SES should be provided – what measure of SES was used to create the median split, and what was the median and range of the cohort reported in this paper?

Segment selection

Why were segments selected for high infant volubility? This seems like a potentially crucial decision that could very much affect the main outcomes of interest. I would hypothesize that segments selected for high infant volubility are more likely to be perceived as including VP. This affects proposition 1 (relative rates of VP versus TT). This could also affect proposition 2, as I would hypothesize that there is more canonical babbling in highly voluble bouts of VP than low-volubility VP. Perhaps I am misunderstanding something important about this selection that the authors can correct or justify, but as it stands I would really like to see some data speaking to this issue to justify the conclusions.

Coder agreement

More needs to be done to justify the use of the VP/TT categorizations with agreement this low. I appreciate the data point the authors provide that conclusions around the higher proportion of VP than TT hold regardless of the set of ratings used, despite low agreement. However, I find it a little disingenuous to refer to this as “the most important matter” and state that “the outcome conclusion would have been the same” when this is only 1 of 4 key hypotheses tested. The authors should explore whether the different sets of ratings would have an impact on the CBR conclusions, as well. Additionally since the authors dichotomize TT and VP for those analysis, I’d be more interested in this binary agreement than the correlation of the 1-6 ratings (I suspect it may look better).

Results

In general the switching between Cohen’s d and mean difference was a little hard for me to track in this whole section. For example the sentence: “Another perspective on the size of the mean difference (.092), however, is that it exceeded the well-known growth in CBR by Age—based on the present data, the CBR growth from 7.5 to 12 months was .081.” was inscrutable to me upon first read. Clarifying that the mean difference here is the difference in CBR between Some VP and No VP would be helpful.

Limitations

It would be nice to acknowledge the lack of video ground truth for coding VP and TT. It seems probable that in many instances of VP, infants are looking at a caregiver as they do it, suggesting a social role even if TT does not occur, which would render VP not strictly endogenous.

Reviewer #3: I was not a reviewer of the original ms, so my comments pertain only to the revised submission. However, I did read the reviews and noted the very extensive changes to the current ms. I can see that the original reviews were highly effective and that the new ms is substantially improved, with a clearer focus as well as more complete results in relation to the issues raised.

I find the basic empirical findings highly convincing – that vocal play is far more frequent than turn-taking, that both types of vocal activity increase over the age range of the study and that canonical babbling is more characteristic of turn-taking than of simple vocal play, despite the fact that the former is far less common. In general, it is apparent that the study – selection of ‘segments’, coding, reliability checks, statistical analysis – was very carefully conducted; I will have nothing to say about those aspects. Instead, I will focus on the abstract, introduction and conclusion and related literature.

Although the abstract seems to be correct as far as it goes and is perfectly clear, a very large proportion of the non-data-based portions of the paper are directed at evolutionary theory. These aspects of the Introduction and Discussion are interesting and worth highlighting – yet there is no mention of them whatsoever in the abstract. This seems to me to be an avoidable error.

I find the literature review somewhat incomplete and the main claim that the paper is designed to challenge – that vocal activity in this period is largely based on imitation, or more precisely, that ‘phonological forms are learned in the first year by infant imitation’ (53f.) – to be something of a straw man (see also 85f, 750ff. and later). More specifically, in a chapter on Precursors to language in Phonological Development (2014, 2nd ed.) Vihman, based on considerable tapping of the literature, presents a view that is not unlike what this paper is meant to show (ll. 755ff.: ‘the possibility that most of the progress in speech-like vocalization of infants that occurs across the first year is the product of…exploratory activity’). And again, reference to Locke’s important 1993 book is good, but one would expect to also see a reference to Vihman et al., 1985, which demonstrated empirically the relationship of infant babbling to their word forms.

More importantly, however, the paper’s evolutionary bias sometimes seems to lead the authors to disregard alternative interpretations. For example, we are told that

Higher rates of canonical syllables during social interaction than during periods without social directivity would suggest a social motivation for producing more advanced protophones. If the idea is on target, we might propose that canonical babbling has been specifically selected to provide a salient signal of developmental progress, especially during social interaction. (ll. 178-181, italics added)

Alternatively, might it not be that infants are stimulated or ‘primed’ by hearing IDS, once canonical babbling is in repertoire, to produce their more advanced, more adult-like babble? Need we jump to the conclusion that they are ‘signalling’ something at this stage (cf. l. 245), when there is little evidence of intentional vocal communication (which tends to emerge only at 13 mos. or so: McCune et al., 1996)? The authors hint at something like this themselves, ll. 490f.

And again, in discussing the unanticipated finding of increases in both vocal play and turn-taking between 7 and 12 months the authors say,

It makes sense to suggest that infants ramp up their vocal activity in all circumstances as time passes, as the possibility of genuine linguistic communication approaches, and as the priority on learning both through exploration and through interaction grows’.

But of course, as the authors are at pains to say themselves, infants (like evolution) cannot anticipate what will be of use later, so that the ‘approach’ of genuine linguistic communication can’t be causal, though it may be caused by this ramping up. Maybe it isn’t infants’ priorities that are changing over this time but rather their mastery of consonant production, their increased supply of well-practiced vocal forms, including syllables, and their sensitivity to a match between the forms they produce and the forms they hear in adult talk, especially IDS, presumably. This is the basis for Vihman’s notion of an ‘articulatory filter’ (1993), a view that the authors appear to echo on ll. 219ff: ‘there may be a tendency of infants to explore their most speech-like utterance capabilities when they are most actively involved in VP, times during which they may be most attentive to their own vocal product, stimulating awareness of their category growth and calibrating their vocal capacities’.(italics added). Most recently, Vihman has suggested that babble functions to create phonological memory, a launchpad for word formation (2022). It would be useful to include such alternative interpretations in the discussion.

Some more specific comments:

- ‘babbling’ is very often used as a short form of ‘canonical babbling’, although it is sometimes used to cover infant vocalizations of the kind Oller and his colleagues term ‘protophones’. This is a systematic source of confusion in the literature and the parenthetical ‘roughly “babbling”’ on l. 76 does not seem helpful (see, for example, the very different use of the same term in McGillion et al., 2017).

- The authors reference ‘effort grunts’ but do not tell us what they are. Note that in this period ‘attention grunts’ have also been observed, although ‘communicative grunts’ appear to arise only beyond age 12 mos.: see McCune et al., 1996, Grünloh et al., 2015. Are these included under protophones or not? They may be the vocal forms that facilitate production of such early-produced word forms as oh.

- use of present perfect where past tense would be appropriate: l. 190, l. 260

Words cited

Grünloh, T. & Liszkowski, U. (2015). Prelinguistic vocalizations distinguish pointing acts. Journal of Child Language, 42, 1312-1336.

McCune, L., Vihman, M. M., Roug-Hellichius, L., Delery, D. B. & Gogate, L. (1996). Grunt communication in human infants (Homo sapiens). Journal of Comparative Psychology, 110, 27-37.

McGillion, M. M., Matthews, D., Herbert, J., Pine, J., Vihman, M. M., Keren-Portnoy, T. & DePaolis, R. A. (2017). What paves the way to conventional language? The predictive value of babble, pointing and SES. Child Development, 88, 156-166.

Vihman, M. M. (1993). Variable paths to early word production. Journal of Phonetics, 21, 61-82.

Vihman, M. M. (2014). Phonological Development: The first two years. (2nd ed.) Malden, MA: Wiley-Blackwell.

Vihman, M. M. (2022). The developmental origins of phonological memory. Psychological Review.

Vihman, M. M., Macken, M.A., Miller, R., Simmons, H. & Miller, J. (1985). From babbling to

7. PLOS authors have the option to publish the peer review history of their article (what does this mean?). If published, this will include your full peer review and any attached files.

Reviewer #2: No

Reviewer #3: No

---

## [Author Response · Author response to Decision Letter 1]

25 Oct 2022

We wish to thank the editor and reviewers for their helpful comments. Please find a point-by-point response to reviewer comments in the attached document.

---

## [Decision Letter · Decision Letter 2]

4 Dec 2022

PONE-D-22-03801R2Perspectives on the origin of language: Infants vocalize most during independent vocal play but produce their most speech-like vocalizations during turn takingPLOS ONE

Dear Dr. Long,

Thank you for submitting your manuscript to PLOS ONE. After careful consideration, we feel that it has merit but does not fully meet PLOS ONE’s publication criteria as it currently stands. Therefore, we invite you to submit a revised version of the manuscript that addresses the points raised during the review process.

ACADEMIC EDITOR:Thank you for your detailed revisions in response to the previous round of reviews. As you will see, both reviewers are satisfied that you have thoroughly addressed their comments. One reviewer has two very minor additional suggestions for you to consider, and so to give you the opportunity to address these as you see fit, I am indicating 'minor revisions' as the decision on the paper. I will then accept your next submission right away.

We look forward to receiving your revised manuscript.

Kind regards,

Marcus Perlman, Ph.D

Academic Editor

PLOS ONE

Journal Requirements:

Reviewers' comments:

Reviewer's Responses to Questions

**Comments to the Author**

1. If the authors have adequately addressed your comments raised in a previous round of review and you feel that this manuscript is now acceptable for publication, you may indicate that here to bypass the “Comments to the Author” section, enter your conflict of interest statement in the “Confidential to Editor” section, and submit your "Accept" recommendation.

Reviewer #2: (No Response)

Reviewer #3: All comments have been addressed

2. Is the manuscript technically sound, and do the data support the conclusions?

Reviewer #2: Yes

Reviewer #3: (No Response)

3. Has the statistical analysis been performed appropriately and rigorously? 

Reviewer #2: Yes

Reviewer #3: (No Response)

4. Have the authors made all data underlying the findings in their manuscript fully available?

Reviewer #2: Yes

Reviewer #3: (No Response)

5. Is the manuscript presented in an intelligible fashion and written in standard English?

Reviewer #2: Yes

Reviewer #3: (No Response)

6. Review Comments to the Author

Reviewer #2: Thank you for the opportunity to review this second revision of the manuscript. The authors have done extensive work to address my concerns, and I am satisfied with the revisions. I have 2 minor comments:

Last sentence of abstract: “The results inform our proposal that the human infant has been naturally selected to explore protophone production and that the exploratory inclination formed a foundation in our hominin ancestors without which language could not have evolved.” The phrase “without which language could not have evolved” seems excessively strong and is not shown by this work. I am in favor of removing it.

Line 375-376: “Volubility could then be thought to drive a tendency for high CBR in TT if the results show that pattern.” I’m not actually sure what “if the results show that pattern” means. Does it mean “if canonical babbling is indeed associated with volubility”? It would be nice to clarify this.

Reviewer #3: (No Response)

7. PLOS authors have the option to publish the peer review history of their article (what does this mean?). If published, this will include your full peer review and any attached files.

Reviewer #2: No

Reviewer #3: No

---

## [Author Response · Author response to Decision Letter 2]

6 Dec 2022

1. We have revised the last sentence of the abstract based on the suggestion of Review 3 to the following: “The results inform our previously published proposal that the human infant has been naturally selected to explore protophone production and that the exploratory inclination in our hominin ancestors formed a foundation for language.”

2. We have also revised lines 375-376 to the following: “Volubility could then be thought to drive a tendency for high CBR in TT if canonical babbling is indeed associated with volubility.”

---

## [Editor Report · Decision Letter 3]

7 Dec 2022

Perspectives on the origin of language: Infants vocalize most during independent vocal play but produce their most speech-like vocalizations during turn taking

PONE-D-22-03801R3

Dear Dr. Long,

We’re pleased to inform you that your manuscript has been judged scientifically suitable for publication and will be formally accepted for publication once it meets all outstanding technical requirements.

Kind regards,

Marcus Perlman, Ph.D

Academic Editor

PLOS ONE
---

## [Editor Report · Acceptance letter]

20 Dec 2022

PONE-D-22-03801R3 

Perspectives on the origin of language: Infants vocalize most during independent vocal play but produce their most speech-like vocalizations during turn taking 

Dear Dr. Long:

I'm pleased to inform you that your manuscript has been deemed suitable for publication in PLOS ONE. Congratulations! Your manuscript is now with our production department. 

Kind regards, 

on behalf of

Dr. Marcus Perlman 

Academic Editor

PLOS ONE